# ADVERSARIALLY ROBUST LEARNING WITH OPTIMAL TRANSPORT REGULARIZED DIVERGENCES

## ABSTRACT

We introduce the $ARMOR_D$ methods as novel approaches to enhancing the adversarial robustness of deep learning models. These methods are based on a new class of optimal-transport-regularized divergences, constructed via an infimal convolution between an information divergence and an optimal-transport (OT) cost. We use these as tools to enhance adversarial robustness by maximizing the expected loss over a neighborhood of distributions, a technique known as distributionally robust optimization (DRO). Viewed as a tool for constructing adversarial samples, our method allows samples to be both transported, according to the OT cost, and re-weighted, according to the information divergence; the addition of a principled and dynamical adversarial re-weighting on top of adversarial sample transport is the key innovation of $ARMOR_D$. We demonstrate the effectiveness of our method on malware detection and image recognition applications and find that it provides significant performance benefits. In malware detection, a discrete (binary) data domain, $ARMOR_D$ improves the robustified accuracy under $rFGSM^{50}$ attack compared to the previous best-performing adversarial training methods by 22 percentage points while simultaneously lowering false negative rate from $4.99\%$ to $2.44\%$.

## 1 INTRODUCTION

Machine learning and specifically deep learning models are known to be vulnerable to adversarial samples: inputs intentionally and meticulously modified by an adversary to evade/mislead the classification model (Papernot et al., 2016; Goodfellow et al., 2014). One common and effective way to enhance a model's robustness against this vulnerability is to include adversarial samples during the training process, known as adversarial training. However, adversarial training is often challenging, as it is hard to maintain the model's performance generalizability while also enhancing its adversarial robustness (Carlini et al., 2019; Zhang et al., 2019). To date, the large body of prominent defense mechanisms for enhancing adversarial robustness includes certifiable approaches Baharlouei et al. (2023); Raghunathan et al. (2018), which can guarantee the absence of adversarial examples misclassified by the model for a specific input, and adversarial training methods Papernot et al. (2017); Madry et al. (2018); Hu et al. (2018); Wang et al. (2020); Zhang et al. (2019; 2020); Dong et al. (2020); Regniez et al. (2021); Bui et al. (2022); Dong et al. (2023) which somehow construct adversarial samples that are employed during training, with Sinha et al. (2018) having aspects of both categories. Despite attractive guarantees, the certifiable approaches often operates on a convex relaxation of the original model rather than the original model and tend to have inferior performance compared to approaches in the latter category (Wang et al., 2020; Athalye et al., 2018).

In the pioneering robust optimization approach Madry et al. (2018) to adversarial training, the loss function $\mathcal{L}_\theta$, depending on parameters $\theta \in \Theta$, is maximized over a metric-space ball centered at the training samples $x_i$, leading to the empirical risk minimization problem

$$\inf_\theta E_{P_n} \left[ \sup_{y:d(x,y)\leq\epsilon} \mathcal{L}_\theta(y) \right], \tag{1}$$

where $P_n = \frac{1}{n}\sum_{i=1}^n \delta_{x_i}$ is the empirical distribution. In Regniez et al. (2021) and Bui et al. (2022) it was recognized that (1) can be expressed as a distributionally robust optimization (DRO) problem over an optimal-transport (OT) neighborhood $\mathcal{U}(P_n) = \{Q : C(Q, P_n) \leq \epsilon\}$ for an appropriate

OT cost $C$; i.e., they noted that (1) $= \inf_\theta \sup_{Q:C(Q,P_n)\leq\epsilon} E_Q[\mathcal{L}_\theta]$. DRO is a general framework for taking a stochastic optimization problem $\inf_\theta E_P[\mathcal{L}_\theta]$ and regularizing (or robustifying) it by maximizing over a neighborhood of distributions, $\mathcal{U}(P)$, around the baseline distribution $P$, leading to the general DRO problem

$$\inf_\theta \sup_{Q\in\mathcal{U}(P)} E_Q[\mathcal{L}_\theta]. \tag{2}$$

This formalizes an uncertainty in the underlying distribution $P$ and can protect against overfitting, leading to better out-of-sample performance; see Rahimian & Mehrotra (2022) for an overview of DRO. For general distribution neighborhoods (2) is an intractable infinite dimensional problem but if $\mathcal{U}$ has the appropriate structure then one can derive tractable finite dimensional reformulations of (2). Prior approaches to the general theory of DRO employ various types of distribution neighborhoods, such as moment constraints Goh & Sim (2010); Delage & Ye (2010); Wiesemann et al. (2014), conditional moment constraints Blanchet et al. (2023), Kullback-Leibler (KL) and $f$-divergence neighborhoods Ben-Tal et al. (2010); Ahmadi-Javid (2012); Hu & Hong (2013); Ben-Tal et al. (2013); Lam (2019), MMD Staib & Jegelka (2019), Wasserstein neighborhoods Mohajerin Esfahani & Kuhn (2018); Shafieezadeh-Abadeh et al. (2019); Wu et al. (2022); Yu-Meng Li & Mao (2022); Gao & Kleywegt (2023), and more general optimal-transport (OT) neighborhoods Blanchet & Murthy (2019).

In the present work we propose a novel class of divergences for comparing probability distributions, which we call the optimal-transport-regularized divergences, that combines features of both OT costs and information-theoretic divergences (such as KL) and use these to define distribution neighborhoods for use in DRO (2). This leads us to propose a novel class of adversarial training methods that simultaneously transport adversarial samples (with general OT cost) and re-weight the adversarial samples according to the information-theoretic divergence. The former feature is shared with the OT-DRO method Bui et al. (2022) (see also the related earlier work Sinha et al. (2018), which used a soft Wasserstein constraint), but the ability of our method to use information from the loss together with the OT cost in order to adversarially re-weight samples in a principled and dynamical manner during training is a qualitatively new feature of our method; this feature follows naturally from our more general DRO framework, which "mixes" information-theoretic and OT divergences via an infimal convolution (see Eq. 3 below). In practice, the adversarial re-weighting causes the optimization algorithm to focus on the samples in each minibatch that are more vulnerable to adversarial perturbation. The DRO-based methods are in contrast to methods which directly modify the loss $\mathcal{L}_\theta$, such as TRADES, Zhang et al. (2019), and MART, Wang et al. (2020). In fact, the two types of techniques can be combined; in Bui et al. (2022) the combination of generalized OT costs with TRADES/MART was shown to lead to further performance gains, beyond either method individually. In this work we focus on evaluating the benefits of the adversarial re-weighting that is inherent to our method; we leave for future work the analysis of our DRO framework in combination with TRADES/MART-style loss modifications.

**Optimal-Transport-Regularized Divergences:** The new divergences that we introduce in this work are defined as an infimal convolution between an optimal transport cost, $C$, and an information divergence, $D$, e.g., an $f$-divergence, $D = D_f$ Liese & Vajda (2006), of which the KL-divergence is one example. More precisely, given an OT cost function $c(x, y)$ and an information divergence, $D$, we define the **OT-regularized divergence**, $D^c$, of a distribution $Q$ with respect to a distribution $P$ by

$$D^c(Q\|P) := \inf_{\eta\in\mathcal{P}(\mathcal{X})}\{D(\eta\|P) + C(\eta, Q)\}, \tag{3}$$

where $\mathcal{P}(\mathcal{X})$ denotes the set of probability distributions on the space $\mathcal{X}$ and the optimal transport cost associated with the cost function $c$ is given by

$$C(\mu, \nu) := \inf_{\pi:\pi_1=\mu,\pi_2=\nu}\int c(x,y)\pi(dxdy) \tag{4}$$

($\pi_i$ denote the marginals of $\pi \in \mathcal{P}(\mathcal{X} \times \mathcal{X})$); the only assumptions we make regarding $c$ are non-negativity, lower semicontinuity, and that $c(x, x) = 0$ for all $x$. Intuitively, one can view (3) as specifying a cost via a two-step procedure for transforming $P$ into $Q$. First, one redistributes the probability-mass in $P$ to form an intermediate distribution $\eta$, paying the cost $D(\eta\|P)$ (we say redistribute because we focus on $D$ that are information divergences, meaning they are computable in terms of the likelihood ratio $d\eta/dP$, though most of our theorems in Appendix A apply more

generally). Second, one performs optimal transport to transform $\eta$ into $Q$, paying the cost $C(\eta, Q)$. The optimal intermediate measure $\eta_*$ determines the final cost $D^c(Q\|P)$. The infimal convolution structure, including the bound $D^c(Q\|P) \leq \min\{D(Q\|P), C(P, Q)\}$, causes $D^c$ to inherit properties from both $D$ and $C$ and allows it to interpolate between these two extremes; see Section 2.2. The OT-regularized divergences are related to the $\Gamma$-divergences defined in Dupuis, Paul & Mao, Yixiang (2022), $(f, \Gamma)$-divergences defined in Birrell et al. (2022) and the IC-$\Gamma$-Rényi divergences from Birrell et al. (2023), but here we utilize optimal transport costs as opposed to integral-probability-metric (IPM) regularization of information divergences. We found that OT-regularization is more naturally suited to adversarial robustness methods than IPM regularization from a mathematical perspective. Also, those prior works focused on the equality of the primal and dual formulas for the divergence, which facilitates applications to GANs; here we focus on adversarial robustness, which requires different techniques.

We use the OT-regularized divergences to define distribution neighborhoods of size $\epsilon > 0$, leading to following DRO problem, which we will employ as a tool for enhancing adversarial robustness

$$\inf_{\theta} \sup_{Q:D^c(Q\|P_n) \leq \epsilon} E_Q[\mathcal{L}_\theta]. \tag{5}$$

The OT-regularized-divergence neighborhoods are qualitatively different from both $f$-divergence and Wasserstein neighborhoods, as they allow for a combination of probability-mass transport and redistribution when forming the perturbed distributions, $Q$. This allows for the support of $Q$ to differ from that of $P_n$ (as in Bui et al. (2022)) and also for the probability of widely separated modes to be re-weighted, something that is not possible with Wasserstein neighborhoods. When viewed as an adversarial training method, we call (5) the $ARMOR_D$ methods, standing for **A**dversarially **R**obust **M**odels with **O**ptimal-Transport-**R**egularized **D**ivergences.

In Section 2 we show how (5) can be converted into a computationally tractable optimization problem and in Section 2.1 we provide a formal solution, thereby clarifying the manner in which our method combines optimal transport and adversarial re-weighting. In Section 2.2 we list a number of properties of the OT-regularized divergences, thus demonstrating that they are well-behaved mathematical objects; precise statements and proofs are found in Appendix A. In Section 3 we test the $ARMOR_D$ methods on MNIST image classification as well as malware classification, where we find it offers significant performance gains.

## 2 OT-REGULARIZED DIVERGENCES: DRO IDENTITY AND PROPERTIES

In general, the DRO problem (5) is an intractable infinite dimensional optimization problem. However, for appropriate choices of $D$ one can derive a finite dimensional reformulation that leads to computationally efficient implementations. In this section we provide a formal derivation of the key identity. For a rigorous proof, and statement of the required assumptions, see Appendix A.2.

Noting that $C$ is jointly convex and assuming that $D$ is convex in its first argument (as is the case when $D$ is an $f$-divergence) one can see from (3) that $D^c$ is convex in its first argument. Therefore the DRO problem is a convex optimization problem and one can compute

$$\sup_{Q:D^c(Q\|P_n) \leq \epsilon} E_Q[\mathcal{L}_\theta] \tag{6}$$

$$= \inf_{\lambda > 0} \{\lambda \epsilon + \sup_{Q \in \mathcal{P}(\mathcal{X})} \{E_Q[\mathcal{L}_\theta] - \lambda D^c(Q\|P_n)\}\} \tag{7}$$

$$= \inf_{\lambda > 0} \{\lambda \epsilon + \sup_{Q, \eta \in \mathcal{P}(\mathcal{X})} \{E_Q[\mathcal{L}_\theta] - \lambda D(\eta\|P_n) - \lambda C(\eta, Q)\}\} \tag{8}$$

$$= \inf_{\lambda > 0} \{\lambda \epsilon + \sup_{\eta \in \mathcal{P}(\mathcal{X})} \{-\lambda D(\eta\|P_n) + \sup_{Q \in \mathcal{P}(\mathcal{X})} \sup_{\pi:\pi_1=\eta, \pi_2=Q} \{E_Q[\mathcal{L}_\theta] - \lambda \int c d\pi\}\}\} \tag{9}$$

$$= \inf_{\lambda > 0} \{\lambda \epsilon + \lambda \sup_{\eta \in \mathcal{P}(\mathcal{X})} \{-D(\eta\|P_n) + \sup_{\pi_x(dy)} \int \int \mathcal{L}_\theta(y)/\lambda - c(x, y) \pi_x(dy) \eta(dx)\}\} \tag{10}$$

$$= \inf_{\lambda > 0} \{\epsilon \lambda + \lambda \sup_{\eta \in \mathcal{P}(\mathcal{X})} \{\int \sup_{y \in \mathcal{X}} \{\lambda^{-1} \mathcal{L}_\theta(y) - c(x, y)\} \eta(dx) - D(\eta\|P_n)\}\}. \tag{11}$$

The equality (7) is obtained using strong duality, lines (8) and (9) are obtained using the definitions (3) and (4) of $D^c$ and $C$ along with properties of suprema and infima, (10) recognizes that the suprema

over $Q$ and $\pi$ can be rewritten as a supremum over probability kernels $\pi_x(dy)$, and finally (11) uses the fact that the supremum over probability kernels achieves the pointwise supremum of the integrand. To this point, the derivation closely follows that of Mohajerin Esfahani & Kuhn (2018) for Wasserstein DRO, as well as the adversarial robustness approach by Bui et al. (2022). Note that effect of the OT cost is to replace the loss $\mathcal{L}_\theta$ with what we call the OT-regularized loss

$$\mathcal{L}^c_{\theta,\lambda}(x) := \sup_{y \in \mathcal{X}} \{\lambda^{-1} \mathcal{L}_\theta(y) - c(x,y)\}, \tag{12}$$

which is known as the $c$-transform in the optimal transport literature; see Definition 5.2 in Villani (2008). The importance of the $c$-transformed loss for Wasserstein DRO is well known; see the references to prior work on Wasserstein and OT-DRO in the introduction. The supremum over $y \in \mathcal{X}$ in (12) can be thought of as selecting an adversarial sample that is paired with each real sample, $x$. We note that our mathematical framework can be used to robustify any empirical risk minimization problem, not only classification, and so our notation does not yet explicitly decompose the variables into sample and label components, though we will do so when applying the method to classification problems in Section 3.

The new ingredient in our OT-regularized-divergence DRO framework is the optimization over $\eta$ in (11). This can be recognized as the convex-conjugate of $\eta \mapsto D(\eta\|P_n)$ and for certain choices of $D$, in particular for the $f$-divergences which we now focus on, this term can be reformulated as a finite dimensional convex optimization problem. Using the generalization of the Gibbs variational principle to $f$-divergences, see Theorem 4.2 in Ben-Tal & Teboulle (2007), one has

$$\sup_{\eta \in \mathcal{P}(\mathcal{X})} \{E_\eta[g] - D_f(\eta\|P)\} = \inf_{\rho \in \mathbb{R}} \{\rho + E_P[f^*(g - \rho)]\}, \tag{13}$$

where $f^*$ is the Legendre transform of $f$. Using this we obtain the following finite-dimensional reformulation of the DRO problem

$$\inf_{\theta \in \Theta} \sup_{Q : D^c_f(Q\|P_n) \leq \epsilon} E_Q[\mathcal{L}_\theta] = \inf_{\lambda > 0, \rho \in \mathbb{R}, \theta \in \Theta} \left\{ \epsilon\lambda + \rho + \lambda \frac{1}{n} \sum_{i=1}^{n} f^*(\mathcal{L}^c_{\theta,\lambda}(x_i) - \rho/\lambda) \right\}. \tag{14}$$

Here we made the change of variables $\rho \to \rho/\lambda$ so that the objective function is jointly convex in $\lambda, \rho$ (see Corollary A.23). Note that the new variables $\lambda, \rho$ simply augment the minimization over model parameters $\theta$ by adding two real variables, which adds very small additional computational cost. The $\lambda$ parameter has the same interpretation as in the OT-DRO based method Bui et al. (2022); it can be viewed as a dynamical OT-cost weight, selected according to the optimization (11), which is tied to the neighborhood size $\epsilon$. This perspective is most apparent in (7). The significance of $\rho$ will be discussed in Section 2.1 below. In Section 3 we will experiment with the KL divergence and the family of $\alpha$-divergences (i.e., $f = f_\alpha$ as in Eq. 29), which we call the $ARMOR_{KL}$ and $ARMOR_\alpha$ methods respectively. An explicit formula for $f^*$ in the case of $\alpha$-divergences is given in (30). In the KL-divergence case the minimization over $\rho$ can be evaluated analytically, yielding

$$\inf_{\theta \in \Theta} \sup_{Q : KL^c(Q\|P_n) \leq \epsilon} E_Q[\mathcal{L}_\theta] = \inf_{\lambda > 0, \theta \in \Theta} \left\{ \epsilon\lambda + \lambda \log\left( \frac{1}{n} \sum_{i=1}^{n} \exp(\mathcal{L}^c_{\theta,\lambda}(x_i)) \right) \right\}. \tag{15}$$

We will refer to either of (14) or (15) as the outer minimization problem and will call (12) the inner maximization problem. While preparing this work a new DRO framework was proposed in Blanchet et al. (2023), employing conditional moment constraints, which was also motivated in part by the desire to combine transport and redistribution costs. Their approach reduces to the $D = D_f$ case of our DRO framework under appropriate assumptions; see their Theorems 4.1, 5.1, and Proposition 5.1 and compare with our Theorem A.22 and Eq. 14-15. Our work is distinguished both mathematically, through the proofs of a number of properties of the OT-regularized divergences that do not have analogues in Blanchet et al. (2023) (see Section 2.2), and through our novel use of (14) and (15) as tools for enhancing adversarial robustness, where we find it leads to substantial performance gains.

## 2.1 INTERPRETING THE OUTER MINIMIZER: ADVERSARIAL SAMPLE WEIGHTS

In this section we give an intuitive interpretation of the minimization over the auxiliary parameters $\lambda, \rho$ in (14); they can be viewed as the computation of optimal adversarial weights for the adversarial samples, where optimality is defined in part by the chosen $f$-divergence. This is a complement to the

inner maximizer (12) which constructs the optimally transported adversarial samples, according to the chosen OT cost function. This interpretation gives insight into the qualitatively novel nature of our method.

Letting $y_i(\lambda)$ be the solution to the inner maximizer (12) with $x = x_i$ and $\lambda_*$ and $\rho_*$ be the optimal scaling and shift parameters for the outer minimizer at a fixed $\theta$ (we suppress the $\theta$-dependence of $y_i$, $\lambda_*$, and $\rho_*$ in the notation) we derive the following reformulation of (14) in Appendix B:

$$\inf_{\lambda > 0, \rho \in \mathbb{R}} \left\{ \epsilon\lambda + \rho + \lambda \frac{1}{n} \sum_{i=1}^{n} f^*(\mathcal{L}_{\theta,\lambda}^c(x_i) - \rho/\lambda) \right\} = E_{Q_{*,\theta}}[\mathcal{L}_\theta], \tag{16}$$

where the optimal adversarial distribution is $Q_{*,\theta} := \sum_{i=1}^{n} p_{*,i} \delta_{y_i(\lambda_*)}$, having optimal adversarial weights

$$p_{*,i} := \frac{1}{n}(f^*)'(\mathcal{L}_{\theta,\lambda_*}^c(x_i) - \rho_*/\lambda_*). \tag{17}$$

This shows that the minimization over $\theta$ in (14) solves the risk minimization problem for the ($\theta$-dependent) optimal adversarial distribution $Q_{*,\theta}$. The optimal adversarial distribution is supported on the optimal adversarial samples $y_i(\lambda_*)$ and the weight of the $i$'th sample is changed from $1/n$ to $p_{*,i}$ (17). To understand the significance of the re-weighting $p_{*,i}$, first recall that $f^*$ is non-decreasing (see Definition A.2 and Corollary A.23), hence $p_{*,i} \geq 0$. In addition, the $p_{*,i}$'s sum to 1 as shown in (90) below. Convexity of $f^*$ implies that $(f^*)'$ is also non-decreasing, hence the $p_{*,i}$'s shift more weight towards the samples where the OT-regularized loss is larger, as would be expected for an adversarial re-weighting. In some cases, such as for the $\alpha$-divergences, $f^*$ is constant on $(-\infty, M)$ for some $M$ ($M = 0$ when $f = f_\alpha$, as seen in Eq. 30). In such cases, samples with $\mathcal{L}_{\theta,\lambda_*}^c(x_i) < \rho_*/\lambda_* + M$ have their weighting changed to 0. Intuitively, one can consider those samples as having sufficiently small OT-regularized loss and hence the method moves its attention away from them to focus on more troublesome samples. These samples are only temporarily ignored; attention may return to them later in the training if their loss moves above the (dynamic) threshold. Part of the task of the outer minimizer is to dynamically determine the optimal threshold for "sufficient smallness", as set by $\rho_*/\lambda_*$. We emphasize that that this threshold changes with $\theta$, as $\lambda_*$ and $\rho_*$ are both $\theta$-dependent.

The ability of the $ARMOR_D$ methods to re-weight adversarial samples in addition to transporting them is the primary innovation of our approach, as compared to the prior OT-DRO based robustness method Bui et al. (2022) or the earlier soft constraint based method Sinha et al. (2018). As we demonstrate in the examples in Section 3 and Appendix C.8, this is a powerful new ingredient and is made possible because our DRO neighborhoods incorporate both information-theoretic and OT components via the infimal convolution (3). Our approach is distinct from the re-weighting method proposed in Guo et al. (2022) for addressing the problem of class imbalance in the training data, which is not an adversarial re-weighting. Our method is also distinct from the approach in Zhang et al. (2020) where modified weights were introduced manually, based on an informal notion of distance to the decision boundary. In contrast, re-weighting in $ARMOR_D$ is determined in a principled manner by the DRO framework, via the choice of $f$ and $c$; it uses information from the OT-regularized loss of each sample during training, along with a dynamic threshold, as seen in (17). In particular the adaptive threshold, which determines which samples the optimizer currently considers "troublesome", is a qualitatively novel feature of our method.

## 2.2 PROPERTIES OF THE OT-REGULARIZED DIVERGENCES

The OT-regularized divergences have many attractive mathematical properties, making them well suited to DRO as well as other statistical learning tasks. We summarize a number of these properties here; see Appendix A for precise statements of the required assumptions along with proofs. Given appropriate assumptions on $D$ and $c$ one has the following:

1. $D^c(\nu\|\mu) \geq 0$ and $D^c(\nu\|\mu) = 0$ if and only if $\nu = \mu$; see Theorem A.7. This divergence property implies that $D^c(\nu\|\mu)$ can be interpreted as measuring the discrepancy between $\nu$ and $\mu$.

2. There exists an optimal intermediate distribution that solves the minimization problem in the definition (3), i.e., there exists $\eta_*$ such that

$$D^c(\nu\|\mu) = D(\eta_*\|\mu) + C(\eta_*, \nu) \tag{18}$$

and this $\eta_*$ is unique under appropriate assumptions. See Theorem A.9.

3. $D^c(\nu\|\mu)$ is convex in $\nu$ (see Lemma A.4). This implies that the DRO neighborhoods $\{Q : D^c(Q\|P_n) \le \epsilon\}$ are convex sets and is also key in the derivation of the DRO identity (11).

4. $D^c(\nu\|\mu)$ is lower semicontinuous in $\nu$ (see Theorem A.11). This property is useful for theoretical purposes and it implies that the DRO neighborhoods $\{Q : D^c(Q\|P_n) \le \epsilon\}$ are closed sets.

5. $D^c$ interpolates between $D$ and $C$ in the following sense: For $r > 0$ define the scaled cost function $c_r = rc$. Then

$$\lim_{r\to 0^+} r^{-1} D^{c_r}(\nu\|\mu) = C(\mu, \nu) \quad \text{(see Theorem A.12)}, \tag{19}$$

$$\lim_{r\to\infty} D^{c_r}(\nu\|\mu) = D(\nu\|\mu) \quad \text{(see Theorem A.13)}. \tag{20}$$

Informally, this property implies that DRO over both $D$ and $C$ neighborhoods can be viewed as special cases of DRO over $D^c$ neighborhoods. More specifically, (19) indicates that when $r$ is sufficiently small, DRO over the neighborhood $\{Q : D^{c_r}(Q\|P_n) \le r\epsilon\}$ is approximately the same as DRO over the neighborhood $\{Q : C(P_n, Q) \le \epsilon\}$. Similarly, (20) indicates that when $r$ is sufficiently large, DRO over the neighborhood $\{Q : D^{c_r}(Q\|P_n) \le \epsilon\}$ is approximately the same as DRO over the neighborhood $\{Q : D(Q\|P_n) \le \epsilon\}$ (see Theorems A.24 and A.25 for precise statements). Therefore if one includes the scale factor $r$ and neighborhood size $\epsilon$ as tunable hyperparameters (as we do in the experiments in Section 3) then the special cases of $C$ and $D$ neighborhoods will be (approximately) explored in the process of tuning an $ARMOR_D$ method.

We note that these properties do not require the distributions to have compact support, except for the DRO interpolation results in Theorems A.24 and A.25.

## 3 EXPERIMENTS

In this section we evaluate the $ARMOR_D$ adversarial robustness methods on two classification problems: MNIST digit classification and malware detection, two common tasks featuring continuous and discrete data, respectively.

**Experimental Setup:** To evaluate the performance of our proposed method, we consider the application of adversarial robustness in two fundamental deep learning tasks: image recognition and malware detection. For the image recognition task we use the MNIST dataset with 50,000 digits in the training and 10,000 in the test set. For the malware detection task, we use a high-dimensional dataset with 22,761 features provided by Al-Dujaili et al. (2018), which includes a total of 54,690 binary encoded malware and benign Windows Portable Executables (PEs) partitioned into training (60%), validation (20%), and test set (20%). Each data point is represented as a 22,761-dimensional binary feature vector denoting the existence of a particular feature in the executable. The target detector models for image and malware data sets were a 4-layer convolutional neural network (CNN) and a 3-layer feed-forward network, respectively, for which the architecture details are given in Appendix C.3. In the binary encoded malware application there is an extra requirement: For the adversarial sample to be functional and preserve malicious malware functionality only bit flips from 0 to 1 are acceptable and not vice versa Al-Dujaili et al. (2018); this gives the problem an inherent asymmetry. Following the guidelines in Carlini et al. (2019), we consider a threat model characterizing the adversary's goal, knowledge, and capabilities detailed in Appendix C.1.

### 3.1 EXPERIMENT 1: ILLUSTRATING THE IMPORTANCE OF ADVERSARIAL RE-WEIGHTING VIA ROBUST IMAGE DETECTION

In this experiment we focus on evaluating the benefit provided by the adversarial sample re-weighting component of the $ARMOR_D$ method alone. Therefore we choose the OT-transport component so that the inner maximizer (12) agrees with the inner maximizer of the Madry et al. (2018) approach, (1), (which we call $PGD$-$AT$). Specifically, in this example we choose

$$c((x, y), (\tilde{x}, \tilde{y})) = \infty 1_{d(x,\tilde{x})>\epsilon} + \infty 1_{y\ne\tilde{y}}, \tag{21}$$

where $x, \tilde{x}$ are samples and $y, \tilde{y}$ are the corresponding labels. This cost allows for the adversarial sample to freely move within the $\epsilon$-ball centered at the original sample, but not outside it, and does not allow for label modification; we let $d$ be the metric induced by the $\infty$-norm.

**Benchmark Methods and Evaluation Metrics:** Following (Bui et al. (2022)), we evaluate the methods against the Projected Gradient Method ($PGD^{200}$) attack and the much stronger and more recent AutoAttack Croce & Hein (2020). In this experiment we are evaluating the $ARMOR_D$'s adversarial re-weighting mechanism alone. Our primary comparison will be the recent OT-DRO based method Bui et al. (2022), called UDR, which like our method can also be used to enhance (1) or any other empirical risk minimization problem. The resulting $UDR$-$PGD$ and $ARMOR_D$-$PGD$ methods are compared on MNIST in Table 1. Following the experiment settings in Bui et al. (2022), all attacks were conducted by a neighborhood size of $0.3$, $\ell_\infty$ neighborhood, and 40 iterations of adversarial training. Implementation details are provided in Appendix C.

**Results:** The best hyperparameters for $ARMOR_D$ were attained via a small grid-search on a parameter space identified in Appendix C.4. To ensure a fair comparison, we closely followed the same settings as in (Bui et al., 2022); see Appendix C.3. We compare the performance of the methods under the attacks $PGD^{200}$ and AutoAttack Croce & Hein (2020) as well as their performance when not under attack (Nat) and report the performance in Table 1. Our proposed method attains higher accuracy than the baseline $PGD$-$AT$ method under both AutoAttack and $PGD^{200}$. The $ARMOR_\alpha$ augmentation of $PGD$ also outperforms $UDR$-$PGD$ on the stronger AutoAttack test. This indicates that the $ARMOR_\alpha$ re-weighting mechanism is an effective tool for enhancing the adversarial robustness of an empirical risk minimization problem. The effects of $ARMOR_\alpha$ can be combined with $UDR$ by modifying the OT-cost function (21) while retaining the sample re-weighing provided by the $f$-divergence component of our method; we intend to explore this for a wider variety of data sets in the future. In the present work, our second example in Section 3.2 explores the use of modified OT-costs within our method.

Table 1: **Enhancing adversarial robustness on MNIST:** Here $ARMOR_\alpha$ uses natural samples alongside the adversarial samples, as described in Appendix C.6. Best metrics are shown in bold font.

| | MNIST | | |
|---|---|---|---|
| **Defense** | AutoAttack | PGD$^{200}$ | Nat |
| $PGD$-$AT$ | 88.9% | 94.0% | 99.4% |
| $UDR$-$PGD$ | 90.0% | **94.3%** | **99.5%** |
| $ARMOR_\alpha$-$PGD$ | **91.70%** | 94.24% | 99.26% |

## 3.2 EXPERIMENT 2: ENHANCING THE ADVERSARIAL ROBUSTNESS OF MALWARE DETECTORS USING A SOFT OT CONSTRAINT AND ADVERSARIAL LABELS

Next we present our results on malware detection, a much higher dimensional and more realistic problem; we closely followed the settings from Al-Dujaili et al. (2018). Appendix C provides the implementation details. In this example we experiment OT cost modifications, which in our approach can be combined with the adversarial re-weighting. We consider two types of OT costs.

**Robust Classification Using Adversarial Samples:** First consider optimal transport cost functions of the form

$$c((x, y), (\tilde{x}, \tilde{y})) = L\|x - \tilde{x}\|^q + \infty 1_{y \neq \tilde{y}} \tag{22}$$

on the space $\mathcal{X} = \mathcal{D} \times \{0, ..., N_c - 1\}$ (i.e., samples in $\mathcal{D} \subset \mathbb{R}^d$ with label from $N_c$ classes). This applies a $q$-Wasserstein cost on the first component (sample) but infinite cost on changing the second component (label); this is a form of soft-constraint, as opposed to the hard-constraint cost (21). The hyperparameter $L > 0$ allows one to choose how much weight is placed on the OT cost, as compared to the information divergence cost in $D^c$. The OT-regularized loss is then

$$\mathcal{L}_{\theta,\lambda}^c(x, y) = \sup_{\tilde{x} \in \mathcal{D}} \{\lambda^{-1}\mathcal{L}_\theta(\tilde{x}, y) - L\|x - \tilde{x}\|^q\}, \tag{23}$$

and corresponds to the construction of a new sample, $\tilde{x}$, adversarial to the original sample $x$ but keeping the original label $y$. We consider the choice of vector norm to be a hyperparameter, selected from $\ell^p$, $p \in [1, \infty]$, and use the cross-entropy loss, $\mathcal{L}_\theta(\tilde{x}, y) = \text{CE}(\phi_\theta(\tilde{x}), y)$ where $\phi_\theta$ is the neural

network (NN) classifier with NN-parameters $\theta$. The adversarial loss (23) can then be used in either of the outer minimizers (14) or (15) (or, more generally, Eq. 11 for some other $D$, provided one can compute its convex conjugate) to obtain an $ARMOR_D$ method. We use the notation $adv_s$ to denote methods that employ adversarial samples constructed via (23).

**Robust Classification Using Adversarial Class Labels and Adversarial Samples:** We will also utilize OT cost functions that allow the class labels to be perturbed in the inner maximizer. To do this we consider the sample space to be $\mathcal{X} = \mathcal{D} \times \mathcal{P}(\{0, ..., N_c - 1\})$ where $\mathcal{P}(\{0, ..., N_c - 1\})$ is the space of probability vectors on the set of labels, with the original class labels mapped to the corresponding one-hot vectors. We relax the term $\infty 1_{y \neq \tilde{y}}$ in (22) to allow for the perturbation of class labels. Allowing for too much label uncertainly will destroy any predictive ability of the classifier, as is also the case with adversarial perturbation of samples, but we find that a small amount improves robustness. To this end, we consider OT cost functions of the form

$$c((x, p), (\tilde{x}, \tilde{p})) = L\|x - \tilde{x}\|^q + Kg_\delta(OT(p, \tilde{p})), \tag{24}$$

where $OT$ is the optimal transport cost (with cost function $1_{i \neq j}$) between the probability vectors $p$ and $\tilde{p}$, i.e., $OT(p, \tilde{p}) = 1 - \sum_{i=1}^{N} \min\{p_i, \tilde{p}_i\}$, and $g_\delta : [0, \delta) \to [0, \infty)$ is increasing, continuous, and satisfies $g_\delta(0) = 0$, $\lim_{z \to \delta^-} g_\delta(z) = \infty$; we then extend the definition via $g_\delta|_{[\delta, \infty)} := \infty$. $K > 0$ is a new cost coefficient hyperparameter and $\delta$ is a new hyperparameter that determines the maximum amount by which the class probabilities can change. More specifically, if the original sample has $p = 1_k$ (i.e., a one-hot vector with a 1 in the $k$'th position, corresponding to the label being $k$) then $OT(p, \tilde{p}) = 1 - \tilde{p}_k$ and so the cost (24) will force the adversarial label to have $\tilde{p}_k > 1 - \delta$. In particular we only use $\delta \in (0, 1/2)$ so that the predicted class is never changed; the class probabilities are only relaxed from being either 0 or 1 to being in $[0, 1]$. Therefore, we do not consider labels to be noisy in the sense discussed in, e.g., Natarajan et al. (2013); Shafieezadeh-Abadeh et al. (2019). We consider this only as a tool to enhance robustness. In our experiments we take $g_\delta(z) = z/(1 - z/\delta)$; note this has a vertical asymptote at $z = \delta$, as required. The inner maximizer with original sample and label being $(x, 1_k)$ is then

$$\mathcal{L}_{\theta, \lambda}^c(x, 1_k) = \sup_{\substack{(\tilde{x}, \tilde{p}) \in \mathcal{X}: \\ \tilde{p}_k > 1 - \delta}} \left\{ \lambda^{-1} \mathcal{L}_\theta(\tilde{x}, \tilde{p}) - L\|x - \tilde{x}\|^q - K \frac{1 - \tilde{p}_k}{1 - (1 - \tilde{p}_k)/\delta} \right\}. \tag{25}$$

We let the baseline loss, $\mathcal{L}_\theta$, be the KL divergence between the adversarial probability vector, $\tilde{p}$, and the classifier output $\phi_\theta(\tilde{x})$ (in the form of a probability vector); note that this is the same as the cross entropy when the label is one-hot but when the labels are relaxed to general probability vectors in the inner maximizer then they differ by the entropy of $\tilde{p}$. We use the notation $adv_{s,l}$ to denote methods that employ both adversarial labels and adversarial samples, constructed via (25).

**Benchmark Methods and Evaluation Metrics:** We consider adversarial training with $r\_FGSM^k$ (Al-Dujaili et al., 2018) and the method proposed by Grosse et al. (2017). We note that Al-Dujaili et al. (2018) proposes several variants for their adversarial training method among which $r\_FGSM^k$ produces the best results. Consistent with Al-Dujaili et al. (2018), we consider three evaluation metrics: accuracy, false negative rate (FNR), and false positive rate (FPR) as well-established evaluation metrics.

**Results:** Table 2 shows the malware experiment results for the non-robust model, benchmark models (the method proposed by Grosse et al. (2017) and $rFGSM^k$), as well as variations of our $ARMOR_D$ method. The best hyperparameters for $ARMOR_D$ were attained via a small grid-search on the parameter space in Appendix C.5. As observed in Table 2, our proposed $ARMOR_\alpha$ ($adv_{s,l}$) achieves the accuracy of $83.31\%$, FNR of $2.44\%$ and FPR of $42.0\%$ against $rFGSM^{50}$ attack outperforming, the benchmark methods and the non-robust model across all three evaluation metrics. $ARMOR_\alpha$ ($adv_{s,l}$) also attains the lowest FNR against $rFGSM^{50}$ and the lowest FNR against Grosse et al.'s attack. We note that, as shown in Table 2, the best performance under attack for Grosse et al. occurs when the adversarial training adopts the same method for inner maximizer. This is aligned with the findings in Al-Dujaili et al. (2018) (see their Table 3). In addition to these results, we also provide experiments to enhance the test generalizability of the malware detector in Appendix D.

Table 2: **Malware adversarial training to enhance performance under attack:** Comparison of the performance of our proposed method in enhancing the robustness on the malware dataset. Hyperparameters were tuned to enhance performance under attack. $adv_s$ denotes the use of adversarial samples constructed via (23) and $adv_{s,l}$ denotes the use of both adversarial samples and labels, as in (25). $nat$ refers to the use of natural samples alongside the adversarial samples, as described in Appendix C.6. $adv^a$ refers to asymmetric methods, as described in Appendix C.7, with only the malicious samples robustified. See Table 5 for results tuned to maximize performance generalizability.

| | $rFGSM^k$ Attack | | | Grosse et al. Attack | | | No Attack | | |
|---|---|---|---|---|---|---|---|---|---|
| **Defense** | Acc | FNR | FPR | Acc | FNR | FPR | Acc | FNR | FPR |
| Non-robust | 14.71% | 77.85% | 98.48% | 33.03% | 99.86% | 8.53% | 92.96% | 5.30% | 10.13% |
| Grosse et al. | 57.36% | 10.96% | 98.91% | **91.08%** | 8.04% | 10.48% | 92.38% | 5.47% | 11.45% |
| $rFGSM^k$ (Al-Dujaili et al.) | 60.79% | 4.99% | 100.00% | 74.74% | 32.39% | 12.59% | 92.83% | 5.20% | 10.66% |
| $ARMOR_{KL}\ (adv_s)$ | 73.86% | 2.80% | 67.61% | 87.59% | 12.50% | 12.26% | 92.58% | 5.44% | 10.94% |
| $ARMOR_{KL}\ (adv_s^a)$ | 69.33% | 5.39% | 95.38% | 85.53% | 16.07% | 11.62% | 92.71% | 5.14% | 11.09% |
| $ARMOR_{KL}\ (adv_s + nat)$ | **84.25%** | 23.33% | **2.28%** | 85.53% | 20.50% | **3.73%** | 92.90% | 5.02% | 10.81% |
| $ARMOR_{KL}\ (adv_s^a + nat)$ | 77.34% | 3.14% | 57.34% | 72.45% | 35.12% | 14.11% | **93.02%** | 5.39% | 9.82% |
| $ARMOR_\alpha\ (adv_s)$ | 83.23% | 5.60% | 36.62% | 89.26% | 9.67% | 12.64% | 92.92% | 5.09% | 10.63% |
| $ARMOR_\alpha\ (adv_s^a)$ | 68.12% | 5.23% | 79.21% | 88.57% | 9.99% | 13.98% | 92.65% | 4.99% | 11.55% |
| $ARMOR_\alpha\ (adv_s + nat)$ | 66.21% | 6.72% | 81.88% | 86.79% | 18.30% | 4.16% | 92.93% | 5.13% | 10.51% |
| $ARMOR_\alpha\ (adv_s^a + nat)$ | 76.20% | 2.86% | 60.99% | 78.68% | 17.66% | 27.82% | 92.38% | **2.71%** | 16.35% |
| $ARMOR_\alpha\ (adv_{s,l})$ | 83.31% | **2.44%** | 42.00% | 71.70% | **1.83%** | 75.33% | 91.08% | 9.00% | **8.78%** |

*Note:* Best metrics are shown in bold font. The numbers for methods that outperform the non-robust model and prior adversarial robustness methods across all three metrics are underlined.

## 4 CONCLUSION

In this work we proposed the $ARMOR_D$ methods for enhancing adversarial robustness of deep learning models. These methods are based on a new class of divergences for comparing probability distributions, the optimal-transport-regularized divergences $D^c$, which are defined as an infimal convolution between an information divergence $D$ (such as KL) and an optimal-transport cost $C$. The key innovation is the principled and dynamical manner in which the method combines transported adversarial samples, along with adversarial re-weighting of the samples via the information divergence. In practice, the adversarial re-weighting focuses the optimization towards improving the performance on the most troublesome adversarial samples. We demonstrated that these new tools have many attractive mathematical properties, making them well suited to applications in statistical learning. The $ARMOR_D$ methods were tested on classification problems representing both continuous (MNIST) and discrete data (malware), where we find that it provides significant performance benefits and outperforms existing methods at enhancing the robustness against adversarial attacks in most tests. For MNIST, we designed the test to isolate the effect of the adversarial sample re-weighting mechanism that is inherent to the $ARMOR_D$ framework. We find that, when used to augment $PGD\text{-}AT$, it increases the performance under AutoAttack by 2.8 percentage points, which is 1.7 points higher than achieved by the recent state-of-the-art OT-based augmentation method Bui et al. (2022). In malware detection, a discrete (binary) data domain, $ARMOR_D$ improves the robustified accuracy under $rFGSM^{50}$ attack compared to the previous best-performing adversarial training methods by 22 percentage points while simultaneously lowering false negative rate from $4.99\%$ to $2.44\%$. These experiments were all done using $ARMOR_D$ where $D$ was an $f$-divergence, however the majority of the rigorous theoretical development we provide in Appendix A applies to a much more general class of $D$'s. Exploring cases beyond $D = D_f$ in the search for new variants of $D^c$ that can be efficiently and effectively applied to adversarial robustness, or to other statistical learning tasks, is an interesting direction for future work. In particular, the Rényi divergences are a natural candidate as their convex conjugate can be computed. Secondly, our method is based on a new general DRO framework of $D^c$ neighborhoods and hence can be used to augment any empirical risk minimization problem. Therefore our work can be used in a manner similar to Bui et al. (2022), which used OT-DRO neighborhoods to obtain enhanced versions of TRADES, Zhang et al. (2019), and MART, Wang et al. (2020); exploring such enhancements using the $D^c$-DRO framework is another promising direction for future work.

## REPRODUCIBILITY STATEMENT

To facilitate reproducibility of the results presented in this paper we include implementation details in Appendix C. Specifically, Appendix C.2 contains pseudocode for the method, Appendix C.3 provides the target networks' structure used for the malware and image applications, Appendix C.5 provides the hyperparameters that yielded the results reported in Tables 3, 2, 4, and 5, Appendix C.6 discusses the implementation of the $adv + nat$ methods, and Appendix C.7 discusses the implementation of the $adv^a$ methods.

## AUTHOR CONTRIBUTIONS

All authors have made equal contribution to this work.

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

## A PROPERTIES OF THE OT-REGULARIZED DIVERGENCES: RIGOROUS STATEMENTS AND PROOFS

In this appendix we rigorously develop the definition and properties of the optimal-transport-regularized divergences that were introduced formally above. Here we will let $\mathcal{X}$ be a Polish space (i.e., a complete separable metric space) with its Borel $\sigma$-algebra (denoted $\mathcal{B}(\mathcal{X})$) and $\mathcal{P}(\mathcal{X})$ will denote the space of Borel probability measures on $\mathcal{X}$. A **pre-divergence** will be a mapping $D : \mathcal{P}(\mathcal{X}) \times \mathcal{P}(\mathcal{X}) \to [0, \infty]$ such that $D(\mu\|\mu) = 0$ for all $\mu \in \mathcal{P}(\mathcal{X})$. We will say that $D$ has the **divergence property** if $D(\mu\|\nu) = 0$ iff $\mu = \nu$. A **cost function** on $\mathcal{X}$ will be a lower semicontinuous (LSC) function $c : \mathcal{X} \times \mathcal{X} \to [0, \infty]$. The associated **optimal-transport (OT) cost** is defined by $C : \mathcal{P}(\mathcal{X}) \times \mathcal{P}(\mathcal{X}) \to [0, \infty]$,

$$C(\mu, \nu) \coloneqq \inf_{\substack{\pi \in \mathcal{P}(\mathcal{X} \times \mathcal{X}): \\ \pi_1 = \mu, \pi_2 = \nu}} \int c\, d\pi \,, \tag{26}$$

where $\pi_i$ denote the marginal distributions. It is a simple exercise to check that if $c(x, x) = 0$ for all $x$ then $C(\mu, \mu) = 0$ for all $\mu$ and if $c(x_1, x_2) = 0$ iff $x_1 = x_2$ then $C(\mu, \nu) = 0$ iff $\mu = \nu$. Also recall that $C$ is convex and is LSC in the product of Prokhorov metric topologies. This follows from Kantorovich duality; see Theorem 5.10 in Villani (2008). All subsequent topological statements regarding probability distributions will refer to the Prokhorov metric topology (i.e., the topology of weak convergence).

Given the above ingredients we now define the class of optimal-transport regularized divergences that are employed in this work.

**Definition A.1.** *Let $D$ be a pre-divergence and $c$ a cost function. The **OT-regularized divergence**, $D^c : \mathcal{P}(\mathcal{X}) \times \mathcal{P}(\mathcal{X}) \to [0, \infty]$, is defined by*

$$D^c(\nu \| \mu) := \inf_{\eta \in \mathcal{P}(\mathcal{X})} \{ D(\eta \| \mu) + C(\eta, \nu) \} . \tag{27}$$

In the main text we frequently referred to $D(\eta \| \mu)$ as an information divergence, meaning it is computable in terms of $d\eta/d\mu$, and the experiments in Section 3 utilized the $f$-divergences, $D = D_f$, which satisfy this property. However our rigorous development in this appendix will be stated more generally. The search for cases beyond $D = D_f$ where $D^c$ can be efficiently applied to adversarial robustness, or to other statistical learning tasks, is an interesting direction for future work. Throughout Section A.1 we provide remarks indicating how the theorems proven here can be applied to OT-regularized $f$-divergences. We use the following definition of $f$-divergences.

**Definition A.2.** *For $a, b \in [-\infty, \infty]$ that satisfy $-\infty \le a < 1 < b \le \infty$ we define $\mathcal{F}_1(a, b)$ to be the set of convex functions $f : (a, b) \to \mathbb{R}$ with $f(1) = 0$. For $f \in \mathcal{F}_1(a, b)$, the corresponding $f$-**divergence** between $\nu, \mu \in \mathcal{P}(\mathcal{X})$ is defined by*

$$D_f(\nu \| \mu) = \begin{cases} E_P[f(d\nu/d\mu)], & \nu \ll \mu \\ \infty, & \nu \not\ll \mu \end{cases} , \tag{28}$$

*where the definition of $f$ in (28) is extended to $[a, b]$ by continuity and is set to $\infty$ on $[a, b]^c$.*

**Remark A.3.** *For certain choices of $f$ one can assign a meaningful finite value to $D_f(\nu \| \mu)$ even when $\nu \not\ll \mu$ Liese & Vajda (2006) but the definition (28) is more convenient for our purposes. That alternative definition agrees with (28) for the choices of $f$ used in the experiments in Section 3.*

In our numerical experiments we use the KL divergence, defined via $f_{KL}(z) = z \log(z)$, and the $\alpha$-divergences, defined via

$$f_\alpha(z) = \frac{z^\alpha - 1}{\alpha(\alpha - 1)}, \quad \alpha > 1 . \tag{29}$$

The Legendre transform of $f_\alpha$ will also be required

$$f_\alpha^*(z) = \alpha^{-1}(\alpha - 1)^{\alpha/(\alpha - 1)} \max\{z, 0\}^{\alpha/(\alpha - 1)} + \frac{1}{\alpha(\alpha - 1)}, \quad \alpha > 1 . \tag{30}$$

### A.1 PROPERTIES OF THE OT-REGULARIZED DIVERGENCES

Here we prove a number of key properties of the OT-regularized divergences.

**Lemma A.4** (Convexity). *Let $D$ be a pre-divergence, $c$ be a cost function, and $\mu \in \mathcal{P}(\mathcal{X})$. If $P \mapsto D(P \| \mu)$ is convex then $P \mapsto D^c(P \| \mu)$ is convex.*

**Remark A.5.** *$f$-divergences satisfy this convexity property. In fact, the map $(Q, P) \to D_f(Q \| P)$ is convex for all $f \in \mathcal{F}_1(a, b)$. This follows from the variational representation of $f$-divergences; see Nguyen et al. (2010), Broniatowski & Keziou (2006) and also Proposition B.1 in Birrell et al. (2022).*

*Proof.* $C$ is convex on $\mathcal{P}(\mathcal{X}) \times \mathcal{P}(\mathcal{X})$ and so $(\eta, \nu) \mapsto D(\eta \| \mu) + C(\eta, \nu)$ is convex. Therefore the infimum over $\eta$ is convex in $\nu$. $\qquad \square$

**Lemma A.6** (Pre-Divergence Property). *Let $D$ be a pre-divergence and $c$ be a cost function that satisfies $c(x, x) = 0$ for all $x$. Then $D^c$ is a pre-divergence.*

*Proof.* We need to show that $D^c(\mu \| \mu) = 0$ for all $\mu \in \mathcal{P}(\mathcal{X})$. To do this we bound the definition by its value at $\eta = \mu$ to obtain

$$0 \le D^c(\mu \| \mu) \le D(\mu \| \mu) + C(\mu, \mu) = 0 , \tag{31}$$

where $C(\mu, \mu) = 0$ follows from the assumption on $c$. $\qquad \square$

**Theorem A.7** (Divergence Property). *Let $D$ be a pre-divergence and $c$ be a cost function that satisfy the following properties.*

    *1. If $D(\mu_n\|\mu) \to 0$ then $\mu_n \to \mu$ weakly.*

    *2. $c(x_1, x_2) = 0$ iff $x_1 = x_2$.*

*Then $D^c$ has the divergence property.*

**Remark A.8.** *In Theorem A.29 below we show that the $f$-divergences satisfy the weak convergence property under mild assumptions and hence this theorem can be applied to OT-regularized $f$-divergences.*

*Proof.* If $\mu = \nu$ then $0 \leq D^c(\nu\|\mu) \leq D(\mu\|\mu) + C(\mu,\nu) = C(\nu,\nu) = 0$. Hence $D^c(\nu\|\mu) = 0$. Conversely if $D^c(\nu\|\mu) = 0$ then there exists a sequence $\eta_n \in \mathcal{P}(\mathcal{X})$ such that $D(\eta_n\|\mu) + C(\eta_n,\nu) \to 0$, i.e., $D(\eta_n\|\mu) \to 0$ and $C(\eta_n,\nu) \to 0$. By the weak convergence property of $D$ we have $\eta_n \to \mu$ weakly. $C$ is LSC, therefore

$$C(\mu,\nu) \leq \lim_{n\to\infty} C(\eta_n,\nu) = 0 \tag{32}$$

and we can conclude that $C(\mu,\nu) = 0$. The assumption on $c$ then implies $\mu = \nu$. $\qquad\square$

Next we provide conditions under which the infimum in (27) has a (unique) solution.

**Theorem A.9.** *Let $D$ be a pre-divergence, $c$ a cost function, and $\mu,\nu \in \mathcal{P}(\mathcal{X})$. If the mapping $P \mapsto D(P\|\mu)$ is LSC and has compact sublevel sets (i.e., $\{P : D(P\|\mu) \leq M\}$ is compact for all $M \in \mathbb{R}$) then there exists $\eta_* \in \mathcal{P}(\mathcal{X})$ such that*

$$D^c(\nu\|\mu) = D(\eta_*\|\mu) + C(\eta_*,\nu). \tag{33}$$

*If $P \mapsto D(P\|\mu)$ is strictly convex on the set where it is finite and $D^c(\nu\|\mu) < \infty$ then this $\eta_*$ is unique.*

**Remark A.10.** *For $f \in \mathcal{F}_1(a,b)$ the $f$-divergences $D_f(\cdot\|\mu)$ are LSC and have compact sublevel sets for all $\mu$, provided that $f^*$ is finite everywhere; see Corollary B.2 and Lemma B.5 in Birrell et al. (2022). If $f$ is strictly convex on $(a,b)$ then $D_f(\cdot\|\mu)$ is strictly convex on the set where it is finite; see Lemma B.6 in Birrell et al. (2022). Therefore Theorem A.9 can be applied to OT-regularized $f$-divergences for appropriate choices of $f$.*

*Proof.* If $D^c(\nu\|\mu) = \infty$ then the definition (27) implies that (33) holds for all $\eta_* \in \mathcal{P}(\mathcal{X})$. Now consider the case where $D^c(\nu\|\mu) < \infty$. Take $\eta_n$ such that $D^c(\nu\|\mu) = \lim_n(D(\eta_n\|\mu) + C(\eta_n,\nu))$. Without loss of generality we can assume that $D(\eta_n\|\mu) \leq D^c(\nu\|\mu) + 1 < \infty$ for all $n$, i.e., $\eta_n$ are all contained in a sublevel set of $D(\cdot\|\mu)$, which is compact by assumption. Therefore there exists a weakly convergent subsequence $\eta_{n_j} \to \eta_*$. Lower semicontinuity of $D(\cdot\|\mu)$ and of $C$ then implies $\liminf_j D(\eta_{n_j}\|\mu) \geq D(\eta_*\|\mu)$ and $\liminf_j C(\eta_{n_j},\nu) \geq C(\eta_*,\nu)$. Therefore

$$D^c(\nu\|\mu) = \lim_j(D(\eta_{n_j}\|\mu) + C(\eta_{n_j},\nu)) \geq D(\eta_*\|\mu) + C(\eta_*,\nu). \tag{34}$$

The reverse inequality is obvious from the definition of $D^c$, hence we can conclude

$$D^c(\nu\|\mu) = D(\eta_*\|\mu) + C(\eta_*,\nu). \tag{35}$$

Now consider the case where $P \mapsto D(P\|\mu)$ is also strictly convex on the set where it is finite. Suppose there exist distinct $\eta_{*,1}, \eta_{*,2} \in \mathcal{P}(\mathcal{X})$ such that

$$D^c(\nu\|\mu) = D(\eta_{*,1}\|\mu) + C(\eta_{*,1},\nu) = D(\eta_{*,2}\|\mu) + C(\eta_{*,2},\nu). \tag{36}$$

Letting $\eta_* = \frac{1}{2}(\eta_{*,1} + \eta_{*,2})$ we can use convexity of $C$ and strict convexity of $D(\cdot\|\mu)$ to compute

$$\begin{aligned}
D^c(\nu\|\mu) \leq{} &D(\eta_*\|\mu) + C(\eta_*,\nu) \tag{37}\\
<{}& \frac{1}{2}D(\eta_{*,1}\|\mu) + \frac{1}{2}D(\eta_{*,2}\|\mu) + \frac{1}{2}C(\eta_{*,1},\nu) + \frac{1}{2}C(\eta_{*,2},\nu)\\
={}& \frac{1}{2}D^c(\nu\|\mu) + \frac{1}{2}D^c(\nu\|\mu) = D^c(\nu\|\mu).
\end{aligned}$$

This is a contradiction, therefore we can conclude the optimizer is unique. $\qquad\square$

Using Theorem A.9 we can prove $D^c(\cdot\|\mu)$ is LSC; see Remark A.10 for the application to OT-regularized $f$-divergences.

**Theorem A.11** (Lower Semicontinuity). *Let $D$ be a pre-divergence, $c$ a cost function, $\mu \in \mathcal{P}(\mathcal{X})$, and assume that $D(\cdot\|\mu)$ is LSC and has compact sublevel sets. Then $\nu \mapsto D^c(\nu\|\mu)$ is LSC.*

*Proof.* Let $\nu_n, \nu \in \mathcal{P}(\mathcal{X})$ with $\nu_n \to \nu$ weakly and define $M := \liminf_n D^c(\nu_n\|\mu)$. If $M = \infty$ then we clearly have $\liminf_n D^c(\nu_n\|\mu) \geq D^c(\nu\|\mu)$ so suppose $M < \infty$. Therefore, fixing $\delta > 0$, there exists $N$ such that for all $n \geq N$ we have $\inf_{j \geq n} D^c(\nu_j\|\mu) < M + \delta$. Hence we can construct a subsequence $j_k$ such that $D^c(\nu_{j_k}\|\mu) < M + \delta$ for all $k$. Theorem A.9 implies that there exists $\eta_{*,n}$ such that

$$D^c(\nu_n\|\mu) = D(\eta_{*,n}\|\mu) + C(\eta_{*,n}, \nu_n) \tag{38}$$

for all $n$, and so

$$M + \delta > D^c(\nu_{j_k}\|\mu) = D(\eta_{*,j_k}\|\mu) + C(\eta_{*,j_k}, \nu_{j_k}) \tag{39}$$

for all $k$. In particular, the $\eta_{*,j_k}$ are contained in the compact sublevel set $\{D(\cdot\|\mu) \leq M + \delta\}$. Therefore there exists a convergent subsequence $\eta_{*,j_{k_\ell}} \to \eta_*$. Lower semicontinuity of $D(\cdot\|\mu)$ and $C$ then implies

$$\begin{aligned}
M + \delta &\geq \liminf_\ell (D(\eta_{*,j_{k_\ell}}\|\mu) + C(\eta_{*,j_{k_\ell}}, \nu_{j_{k_\ell}})) \tag{40}\\
&\geq \liminf_\ell D(\eta_{*,j_{k_\ell}}\|\mu) + \liminf_\ell C(\eta_{*,j_{k_\ell}}, \nu_{j_{k_\ell}}))\\
&\geq D(\eta_*\|\mu) + C(\eta_*, \nu)\\
&\geq D^c(\nu\|\mu) \,.
\end{aligned}$$

Taking $\delta \to 0^+$ and recalling the definition of $M$ completes the proof. $\square$

Finally, we prove a pair of results showing that $D^c$ reduces to either $D$ or $C$ in certain limits. Therefore one can think of $D^c$ as a type of interpolation between $D$ and $C$. To apply these theorems to the case $D = D_f$, see Remarks A.8 and A.10.

**Theorem A.12** (Interpolation). *Let $D$ be a pre-divergence, $c$ be a cost function, and $\mu, \nu \in \mathcal{P}(\mathcal{X})$ that satisfy the following.*

1. *The mapping $P \mapsto D(P\|\mu)$ is LSC and has compact sublevel sets.*

2. *$D(\mu_n\|\mu) \to 0$ implies $\mu_n \to \mu$ weakly.*

*For $r > 0$ define the cost function $c_r = rc$. Then*

$$\lim_{r \to 0^+} r^{-1} D^{c_r}(\nu\|\mu) = C(\mu, \nu) \ \text{ for all } \mu, \nu \in \mathcal{P}(\mathcal{X}). \tag{41}$$

*Proof.* From the definitions we have

$$r^{-1} D^{c_r}(\nu\|\mu) = \inf_{\eta \in \mathcal{P}(\mathcal{X})} \{r^{-1} D(\eta\|\mu) + C(\eta, \nu)\} \tag{42}$$

and the right-hand side is non-increasing in $r$. Therefore for $r_n \searrow 0$ we have

$$\lim_n r_n^{-1} D^{c_{r_n}}(\nu\|\mu) = \sup_n r_n^{-1} D^{c_{r_n}}(\nu\|\mu) \leq C(\mu, \nu) \,, \tag{43}$$

where the inequality comes from bounding (27) by its value at $\eta = \mu$. We will show that the assumption that this inequality is strict leads to a contradiction, which will complete the proof. If the inequality is strict then $D^{c_{r_n}}(\nu\|\mu) < \infty$ for all $n$ and Theorem A.9 implies the existence of $\eta_{*,n}$ such that

$$D(\eta_{*,n}\|\mu) \leq D(\eta_{*,n}\|\mu) + r_n C(\eta_{*,n}, \nu) = D^{c_{r_n}}(\nu\|\mu) \leq r_n \sup_m r_m^{-1} D^{c_{r_m}}(\nu\|\mu) < \infty \,. \tag{44}$$

Taking $n \to \infty$ we see that $D(\eta_{*,n}\|\mu) \to 0$ and therefore $\eta_{*,n} \to \mu$ weakly. $C$ is LSC, therefore $\liminf_n C(\eta_{*,n}, \nu) \geq C(\mu, \nu)$. Combining these we have

$$C(\mu, \nu) > \sup_n r_n^{-1} D^{c_{r_n}}(\nu\|\mu) \geq \liminf_n (r_n^{-1} D(\eta_{*,n}\|\mu) + C(\eta_{*,n}, \nu)) \tag{45}$$

$$\geq \liminf_n C(\eta_{*,n}, \nu) \geq C(\mu, \nu) \,. \tag{46}$$

This is a contradiction, hence the proof is complete. $\square$

**Theorem A.13** (Interpolation). *Let $D$ be a pre-divergence, $c$ be a cost function, and $\mu, \nu \in \mathcal{P}(\mathcal{X})$ that satisfy the following.*

    *1. The mapping $P \mapsto D(P\|\mu)$ is LSC and has compact sublevel sets.*

    *2. $c(x_1, x_2) = 0$ iff $x_1 = x_2$.*

*For $r > 0$ define the cost function $c_r = rc$. Then*

$$\lim_{r \to \infty} D^{c_r}(\nu\|\mu) = D(\nu\|\mu) \ \text{ for all } \mu, \nu \in \mathcal{P}(\mathcal{X}). \tag{47}$$

*Proof.* From the definitions we have

$$D^{c_r}(\nu\|\mu) = \inf_{\eta \in \mathcal{P}(\mathcal{X})} \{D(\eta\|\mu) + rC(\eta, \nu)\} \leq D(\nu\|\mu) \tag{48}$$

and the left-hand side is non-decreasing in $r$. Therefore for $r_n \nearrow \infty$ we have

$$\lim_{n \to \infty} D^{c_{r_n}}(\nu\|\mu) = \sup_n D^{c_{r_n}}(\nu\|\mu) \leq D(\nu\|\mu). \tag{49}$$

Assuming this inequality is strict will lead to a contradiction, thus completing the proof. Suppose that $\sup_n D^{c_{r_n}}(\nu\|\mu) < D(\nu\|\mu)$. Theorem A.9 implies the existence of $\eta_{*,n}$ such that

$$D^{c_{r_n}}(\nu\|\mu) = D(\eta_{*,n}\|\mu) + r_n C(\eta_{*,n}, \nu). \tag{50}$$

In particular, $\sup_n D(\eta_{*,n}\|\mu) \leq \sup_n D^{c_{r_n}}(\nu\|\mu) < \infty$ and therefore $\eta_{*,n}$ all lie in a sublevel set of $D(\cdot\|\mu)$, which is compact. Hence there exists a weakly convergent subsequence $\eta_{*,n_j} \to \eta_*$. Next we show that $\eta_* = \nu$. To do this, note that

$$\infty > \sup_n D^{c_{r_n}}(\nu\|\mu) \geq \sup_n r_n C(\eta_{*,n}, \nu) \tag{51}$$

and therefore $\lim_n C(\eta_{*,n}, \nu) = 0$. Lower semicontinuity implies $0 = \lim_j C(\eta_{*,n_j}, \nu) \geq C(\eta_*, \nu) \geq 0$ and so $C(\eta_*, \nu) = 0$. The cost function has the property $c(x_1, x_2) = 0$ iff $x_1 = x_2$, hence we can conclude that $\eta_* = \nu$. To complete the proof we can use the lower semicontinuity of $D(\cdot\|\mu)$ to compute

$$D(\nu\|\mu) > \sup_n D^{c_{r_n}}(\nu\|\mu) \geq \liminf_j D(\eta_{*,n_j}\|\mu) \geq D(\eta_*\|\mu) = D(\nu\|\mu). \tag{52}$$

This is a contradiction and so the proof is complete. $\qquad\square$

## A.2    DRO USING OT-REGULARIZED DIVERGENCES

In this section we provide rigorous proofs for the key identities that transform the DRO problem over OT-regularized-divergence neighborhoods into a computationally tractable form. This will involve the construction of regularized loss functions, as defined below.

**Definition A.14.** *Given a loss function $\mathcal{L} : \mathcal{X} \to [-\infty, \infty]$ we define the corresponding family of* **OT-regularized losses** *by*

$$\mathcal{L}^c_\lambda(x) := \sup_{y \in \mathcal{X}} \{\mathcal{L}(y)/\lambda - c(x, y)\}, \ \ \lambda > 0, \tag{53}$$

*where we employ the convention $\infty - \infty := -\infty$. $\mathcal{L}^c_\lambda$ is known as the c-transform in the optimal transport literature; see Definition 5.2 in Villani (2008).*

**Remark A.15.** *From a mathematical perspective, the convention $\infty - \infty := -\infty$ is motivated by the proof of Theorem A.18 below. It also coincides with the behavior one intuitively wants based on viewing the maximization in (53) as the construction of a new sample $y$ that is adversarial to the original sample $x$. If the transport cost $c(x, y) = \infty$ then one should view $y$ as impossible to reach when starting from $x$ and so $y$ should not be a valid adversarial sample to pair with $x$, even if $\mathcal{L}(y) = \infty$. Therefore such $y$'s should be excluded from the maximization in (53); mathematically, this corresponds to defining $\infty - \infty := -\infty$.*

In the main text we use DRO as a tool for enhancing adversarial robustness, and there we consider distribution neighborhoods of the form $\{Q : D^c(Q\|P_n) \le \epsilon\}$, where the baseline distribution is an empirical distribution $P_n$. However, it can be useful to have a proof of the DRO identity for the neighborhoods $\{Q : D^c(Q\|P) \le \epsilon\}$ with a general baseline distribution $P$ and so we study this more general problem below. A key tool will be the following interchangeability result, which has previously been used in Wasserstein and OT DRO; see the discussion in Zhang et al. (2022). For completeness we provide a proof of the version employed in this work; our proof mimics the strategy used for the more general result stated in Zhang et al. (2022). Below we will use the notation $\mathcal{M}_\mu$ for the completion of a $\sigma$-algebra, $\mathcal{M}$, with respect to a measure $\mu$, we will denote the completion of $\mu$ by $\overline{\mu}$, and $\mathcal{M}_*$ will denote the $\sigma$-algebra of universally measurable sets (with respect to $\mathcal{M}$).

**Lemma A.16** (Interchangeability). *Let $\mu \in \mathcal{P}(\mathcal{X})$ and $\phi : \mathcal{X} \times \mathcal{X} \to [-\infty, \infty]$ be measurable. Then $x \mapsto \sup_{y \in \mathcal{X}} \phi(x, y)$ is a $\mathcal{B}(\mathcal{X})_*$-measurable function and*

$$\sup_{\pi \in \mathcal{P}(\mathcal{X} \times \mathcal{X}): \pi_1 = \mu} E_\pi[\phi] = \int \sup_{y \in \mathcal{X}} \phi(x, y) \overline{\mu}(dx). \tag{54}$$

**Remark A.17.** *In (54) we use the convention $\infty - \infty := -\infty$ to ensure all integrals therein are defined, though when using this result in the proof of Theorem A.18 below we will have further assumptions that guarantee all integrals are defined without relying on any such convention.*

*Proof.* Define $\Phi = \sup_{y \in \mathcal{X}} \phi(\cdot, y)$. For $a \in \mathbb{R}$ we have

$$\{x : \Phi(x) > a\} = \{x : \exists y, \phi(x, y) > a\}, \tag{55}$$

which is the projection of the measurable set $\phi^{-1}((a, \infty])$ onto its first component. Therefore the measurable projection theorem (see, e.g., Proposition 8.4.4 in Cohn (2013)) implies $\{x : \Phi(x) > a\}$ is $\mathcal{B}(\mathcal{X})_*$-measurable. The rays $(a, \infty]$ for $a \in \mathbb{R}$ generate the $\sigma$-algebra on $[-\infty, \infty]$, hence $\Phi$ is universally measurable as claimed.

To prove (54), first suppose $\int \Phi^- d\overline{\mu} < \infty$, where $\Phi^-$ denotes the negative part of $\Phi$. Define $\Phi_n = \min\{n, \Phi - 1/n\}$ and note that $\min\{0, \Phi - 1\} \le \Phi_n < \Phi$, and $\Phi_n \nearrow \Phi$. The $\Phi_n$ are universally measurable, therefore $C_n := \{(x, y) \in \mathcal{X} \times \mathcal{X} : \phi(x, y) > \Phi_n(x)\}$ are $\mathcal{B}(\mathcal{X})_\mu \bigotimes \mathcal{B}(\mathcal{X})$-measurable. For every $x \in \mathcal{X}$ we have $\Phi_n(x) < \Phi(x) = \sup_{y \in \mathcal{X}} \phi(x, y)$, hence there exists $y \in \mathcal{X}$ such that $(x, y) \in C_n$. Therefore the projection of $C_n$ onto its first component equals $\mathcal{X}$. The measurable selection theorem, Corollary 8.5.4 in Cohn (2013), then implies that there exists $T_n : \mathcal{X} \to \mathcal{X}$ that is $((\mathcal{B}(\mathcal{X})_\mu)_*, \mathcal{B}(\mathcal{X}))$-measurable such that the graph of $T_n$ is contained in $C_n$. Using the result of Cohn Ex. 8.4.2(b) we have $(\mathcal{B}(\mathcal{X})_\mu)_* = \mathcal{B}(\mathcal{X})_\mu$, therefore $T_n$ is $(\mathcal{B}(\mathcal{X})_\mu, \mathcal{B}(\mathcal{X}))$-measurable. The map $\psi : x \mapsto (x, T_n(x))$ is $(\mathcal{B}(\mathcal{X})_\mu, \mathcal{B}(\mathcal{X}) \bigotimes \mathcal{B}(\mathcal{X}))$-measurable and the pushforward measure $\psi_\# \overline{\mu} \in P(\mathcal{X} \times \mathcal{X})$ satisfies $(\psi_\# \overline{\mu})_1 = \mu$, therefore

$$\sup_{\pi \in \mathcal{P}(\mathcal{X} \times \mathcal{X}): \pi_1 = \mu} E_\pi[\phi] \ge E_{\psi_\# \overline{\mu}}[\phi] = \int \phi(x, T_n(x)) \overline{\mu}(dx) \ge \int \Phi_n d\overline{\mu}, \tag{56}$$

where in the last inequality we used the fact that the graph of $T_n$ is contained in $C_n$. We have the lower bound $\Phi_n \ge -\Phi^- - 1 \in L^1(\overline{\mu})$ and therefore we can use the monotone convergence theorem to obtain

$$\sup_{\pi \in \mathcal{P}(\mathcal{X} \times \mathcal{X}): \pi_1 = \mu} E_\pi[\phi] \ge \lim_{n \to \infty} \int \Phi_n d\overline{\mu} = \int \Phi d\overline{\mu}. \tag{57}$$

This also trivially holds if $\int \Phi^- d\overline{\mu} = \infty$ due to our convention $\infty - \infty := -\infty$. The reverse inequality follows easily from the bound $\Phi(x) \ge \phi(x, y)$ for all $x$ and $y$, together with the fact that there exists a $\mathcal{B}(\mathcal{X})$-measurable function that equals $\Phi$ $\overline{\mu}$-a.s. $\square$

Now we derive a formula that relates the convex conjugate of $D^c(\cdot\|P)$ to the convex conjugate of $D(\cdot\|P)$. This is a useful result in its own right and is a key ingredient in solving the DRO problem.

**Theorem A.18.** *Suppose we have the following:*

1. *A measurable function $\mathcal{L} : \mathcal{X} \to [-\infty, \infty]$ that is bounded below or is bounded above.*

2. *A distribution $P \in \mathcal{P}(\mathcal{X})$.*

3. *A pre-divergence, $D$, such that $D(\cdot\|P)$ is convex.*

4. *A cost function, $c$, that satisfies $c(x,x)=0$ for all $x \in \mathcal{X}$.*

*Then for $\lambda > 0$ we have*

$$\sup_{\substack{Q \in \mathcal{P}(\mathcal{X}): \\ D^c(Q\|P)<\infty}} \{E_Q[\mathcal{L}] - \lambda D^c(Q\|P)\} = \sup_{\substack{Q \in \mathcal{P}(\mathcal{X}): \\ D(Q\|P)<\infty}} \{E_Q[\lambda \mathcal{L}_\lambda^c] - \lambda D(Q\|P)\}, \qquad (58)$$

*where $\mathcal{L}_\lambda^c$ (defined in Eq. 53) is a universally measurable function.*

**Remark A.19.** *In Theorem A.18 and in the following, when it is convenient for simplifying notation we use the same symbol to denote a probability measure and its completion, as the correct interpretation is easily discovered by examining the measurably of the integrand. When needed for clarity, we will again use $\overline{Q}$ to denote the completion of $Q \in \mathcal{P}(\mathcal{X})$.*

*Proof.* Universal measurability of $\mathcal{L}_\lambda^c$ follows from the interchangability result, Lemma A.16. To prove (58), first suppose that $\mathcal{L}$ is bounded above. Using the definitions of $D^c$ and $C$ we can compute

$$\sup_{Q \in \mathcal{P}(\mathcal{X}): D^c(Q\|P)<\infty} \{E_Q[\mathcal{L}] - \lambda D^c(Q\|P)\} \qquad (59)$$

$$= \sup_{Q \in \mathcal{P}(\mathcal{X})} \left\{ E_Q[\mathcal{L}] - \lambda \inf_{\eta \in \mathcal{P}(\mathcal{X})} \{D(\eta\|P) + C(\eta, Q)\} \right\}$$

$$= \lambda \sup_{\eta \in \mathcal{P}(\mathcal{X})} \left\{ \sup_{Q \in \mathcal{P}(\mathcal{X})} \{E_Q[\mathcal{L}/\lambda] - C(\eta, Q)\} - D(\eta\|P) \right\}$$

$$= \lambda \sup_{\eta \in \mathcal{P}(\mathcal{X}): D(\eta\|P)<\infty} \left\{ \sup_{Q \in \mathcal{P}(\mathcal{X})} \left\{ \sup_{\pi: \pi_1=\eta, \pi_2=Q} \left\{ \int \lambda^{-1}\mathcal{L}(y) - c(x,y)\pi(dxdy) \right\} \right\} - D(\eta\|P) \right\}$$

$$= \lambda \sup_{\eta \in \mathcal{P}(\mathcal{X}): D(\eta\|P)<\infty} \left\{ \sup_{\pi: \pi_1=\eta} \left\{ \int \lambda^{-1}\mathcal{L}(y) - c(x,y)\pi(dxdy) \right\} - D(\eta\|P) \right\}$$

$$= \lambda \sup_{\eta \in \mathcal{P}(\mathcal{X}): D(\eta\|P)<\infty} \left\{ \int \sup_{y \in \mathcal{X}} \{\lambda^{-1}\mathcal{L}(y) - c(x,y)\}\eta(dx) - D(\eta\|P) \right\}, \qquad (60)$$

where we used the interchangability result, Lemma A.16, to obtain the last line. The assumption that $\mathcal{L}$ is bounded above, and hence $E_Q[\mathcal{L}] \in [-\infty, \infty)$ for all $Q$, ensured that $\infty - \infty$ was not encountered in (59)-(60). Recalling the definition (53) this completes the proof when $\mathcal{L}$ is bounded above.

Now suppose $\mathcal{L}$ is bounded below. Define $\mathcal{L}_n(x) := \min\{\mathcal{L}(x), n\}$, $n \in \mathbb{Z}^+$. These are bounded below uniformly in $n$ and so $\sup_n E_Q[\mathcal{L}_n] = E_Q[\mathcal{L}]$ for all $Q$ by the monotone convergence theorem. The $\mathcal{L}_n$ are bounded above, hence we can use (59)-(60) to obtain

$$\sup_{Q \in \mathcal{P}(\mathcal{X}): D^c(Q\|P)<\infty} \{E_Q[\mathcal{L}] - \lambda D^c(Q\|P)\} \qquad (61)$$

$$= \sup_n \sup_{Q \in \mathcal{P}(\mathcal{X}): D^c(Q\|P)<\infty} \{E_Q[\mathcal{L}_n] - \lambda D^c(Q\|P)\}$$

$$= \sup_n \sup_{Q \in \mathcal{P}(\mathcal{X}): D(Q\|P)<\infty} \left\{ \lambda \int (\mathcal{L}_n)_\lambda^c(x)Q(dx) - \lambda D(Q\|P) \right\}$$

$$= \sup_{Q \in \mathcal{P}(\mathcal{X}): D(Q\|P)<\infty} \left\{ \lambda \sup_n \int (\mathcal{L}_n)_\lambda^c(x)Q(dx) - \lambda D(Q\|P) \right\}$$

$$= \sup_{Q \in \mathcal{P}(\mathcal{X}): D(Q\|P)<\infty} \{\lambda E_Q[\mathcal{L}_\lambda^c] - \lambda D(Q\|P)\},$$

where, noting that the functions $(\mathcal{L}_n)_\lambda^c$ are bounded below uniformly in $n$, we again used the monotone convergence theorem in the final equality. We emphasize that the convention $\infty - \infty = -\infty$ is needed to justify the computation $\sup_n (\mathcal{L}_n)_\lambda^c(x) = \sup_y \{\sup_n \mathcal{L}_n(y)/\lambda - c(x,y)\} = \mathcal{L}_\lambda^c(x)$ for all $x$. This proves the claim when $\mathcal{L}$ is bounded below and so the proof is complete. $\square$

In particular, when $D = D_f$ is an $f$-divergence we can further evaluate the convex conjugate of $D_f$ to obtain a formula that only involves expectations with respect to $P$.

**Corollary A.20.** *Suppose we have the following:*

1. *A measurable function $\mathcal{L} : \mathcal{X} \to [-\infty, \infty]$ that is bounded below or is bounded above.*

2. *A distribution $P \in \mathcal{P}(\mathcal{X})$ such that $\mathcal{L}^- \in L^1(P)$, where $\mathcal{L}^-$ denotes the negative part of $\mathcal{L}$.*

3. *$f \in \mathcal{F}_1(a, b)$ with $a \geq 0$.*

4. *A cost function, c, that satisfies $c(x, x) = 0$ for all $x \in \mathcal{X}$.*

*Then for $\lambda > 0$ we have*

$$\sup_{Q \in \mathcal{P}(\mathcal{X}) : D_f^c(Q\|P) < \infty} \{E_Q[\mathcal{L}] - \lambda D_f^c(Q\|P)\} = \lambda \inf_{\rho \in \mathbb{R}} \{\rho + E_P[f^*(\mathcal{L}_\lambda^c - \rho)]\}, \qquad (62)$$

*where the definition of $f^*$ is extended by $f^*(\pm\infty) := \infty$.*

*Proof.* $D_f$ is a pre-divergence and $D_f(\cdot\|P)$ is convex, hence Theorem A.18 gives

$$\sup_{Q \in \mathcal{P}(\mathcal{X}) : D_f^c(Q\|P) < \infty} \{E_Q[\mathcal{L}] - \lambda D_f^c(Q\|P)\} = \lambda \sup_{Q \in \mathcal{P}(\mathcal{X}) : D_f(Q\|P) < \infty} \{E_{\overline{Q}}[\mathcal{L}_\lambda^c] - D_f(Q\|P)\}$$
$$(63)$$

for all $\lambda > 0$. We have $\mathcal{L}_\lambda^c(x) \geq \mathcal{L}(x)/\lambda$ and so $(\mathcal{L}_\lambda^c)^- \leq \mathcal{L}^-/\lambda \in L^1(P)$. Therefore $(\mathcal{L}_\lambda^c)^- \in L^1(P)$ and we can employ the Gibbs variational principle for $f$-divergences (see Theorem 4.2 in Ben-Tal & Teboulle (2007)) to compute

$$\sup_{Q : D_f(Q\|P) < \infty} \{E_{\overline{Q}}[\mathcal{L}_\lambda^c] - D_f(Q\|P)\} = \inf_{\rho \in \mathbb{R}} \{\rho + E_{\overline{P}}[f^*(\mathcal{L}_\lambda^c - \rho)]\}. \qquad (64)$$

We revert to explicit completion notation here to clarify a technical point. Theorem 4.2 from Ben-Tal & Teboulle (2007) assumes measurability of the integrand and not universal measurability. However one can easily prove that (64) still follows by first replacing $\mathcal{L}_\lambda^c$ with a $\mathcal{B}(\mathcal{X})$-measurable function that agrees with it $\overline{P}$-a.s. and then using the fact that $D_f(Q\|P) < \infty$ implies $Q \ll P$; see Definition (A.2). Also, the result in Ben-Tal & Teboulle (2007) assumes the integrand on the left-hand side of (64) is in $L^1(P)$ but the case where the positive part is not integrable is easily checked to yield infinity on both sides of the identity. Combining (64) and (63) completes the proof. $\qquad\square$

Before proceeding to the DRO problem we need a lemma regarding the finiteness of expectations as the distribution ranges over an OT-regularized-divergence neighborhood.

**Lemma A.21.** *Suppose we have the following:*

1. *A measurable function $\mathcal{L} : \mathcal{X} \to [-\infty, \infty]$.*

2. *A distribution $P \in \mathcal{P}(\mathcal{X})$*

3. *A pre-divergence, $D$, such that $D(\cdot\|P)$ is convex.*

4. *A cost function, c, that satisfies $c(x, x) = 0$ for all $x \in \mathcal{X}$.*

*Suppose there exists $\epsilon > 0$ such that $\mathcal{L} \in L^1(Q)$ for all $Q \in \mathcal{P}(\mathcal{X})$ that satisfy $D^c(Q\|P) \leq \epsilon$. Then $\mathcal{L} \in L^1(Q)$ for all $Q$ that satisfy $D^c(Q\|P) < \infty$.*

*Proof.* The assumptions imply that $D^c$ is a pre-divergence (see Lemma A.6) and $D^c(\cdot\|P)$ is convex (see Lemma A.4). Take any $Q$ with $D^c(Q\|P) < \infty$ and define $Q_t = tQ + (1 - t)P$ for $t \in (0, 1)$. By convexity and the pre-divergence property we have $D^c(Q_t\|P) \leq tD^c(Q\|P)$. We assumed $D^c(Q\|P) < \infty$, hence there exists $t \in (0, 1)$ with $D^c(Q_t\|P) \leq \epsilon$. This implies $\mathcal{L} \in L^1(Q_t)$ and so $\infty > E_{Q_t}[|\mathcal{L}|] = tE_Q[|\mathcal{L}|] + (1 - t)E_P[|\mathcal{L}|]$. We have $t \in (0, 1)$, therefore we can conclude $E_Q[|\mathcal{L}|] < \infty$ as claimed. $\qquad\square$

We are now ready to consider the DRO problem for general $P$. We also allow for an explicit $D^c$ penalty term, in addition to maximizing over the distribution neighborhood, though we do not employ such a penalty term in the experiments in Section 3.

**Theorem A.22.** *Suppose we have the following:*

1. *A measurable function $\mathcal{L} : \mathcal{X} \to [-\infty, \infty]$ that is bounded below or is bounded above.*

2. *A distribution $P \in \mathcal{P}(\mathcal{X})$ such that $\mathcal{L}^- \in L^1(P)$.*

3. *A pre-divergence, $D$, such that $D(\cdot \| P)$ is convex.*

4. *A cost function, $c$, that satisfies $c(x, x) = 0$ for all $x \in \mathcal{X}$.*

*Then for $\epsilon > 0$, $\kappa \geq 0$ we have*

$$\sup_{Q : D^c(Q\|P) \leq \epsilon} \{E_Q[\mathcal{L}] - \kappa D^c(Q\|P)\} \tag{65}$$
$$= \inf_{\lambda > 0} \{\lambda \epsilon + (\lambda + \kappa) \sup_{Q \in \mathcal{P}(\mathcal{X}) : D(Q\|P) < \infty} \{E_Q[\mathcal{L}^c_{\lambda+\kappa}] - D(Q\|P)\}\} .$$

*Proof.* We will show that

$$\sup_{Q : D^c(Q\|P) \leq \epsilon} \{E_Q[\mathcal{L}] - \kappa D^c(Q\|P)\} = \inf_{\lambda > 0} \{\lambda \epsilon + \sup_{Q : D^c(Q\|P) < \infty} \{E_Q[\mathcal{L}] - (\lambda + \kappa) D^c(Q\|P)\}\} .$$
$$\tag{66}$$

Combining this with the result of Theorem A.18 will then complete the proof. If there exists $Q$ such that $D^c(Q\|P) \leq \epsilon$ and $E_Q[\mathcal{L}^+] = \infty$ then it is straightforward to see that both sides of (66) equal $\infty$. Therefore it suffices to consider the case where $E_Q[\mathcal{L}^+] < \infty$ for all $Q$ satisfying $D^c(Q\|P) \leq \epsilon$. Applying Lemma A.21 to $\mathcal{L}^+$ then implies $\mathcal{L}^+ \in L^1(Q)$ for all $Q$ satisfying $D^c(Q\|P) < \infty$. Therefore the $F : Q \mapsto E_Q[\mathcal{L}] - \kappa D^c(Q\|P)$ is a concave map from $\{Q : D^c(Q\|P) < \infty\}$ to $[-\infty, \infty)$ and $Q \mapsto D^c(Q\|P)$ is a convex constraint. $P$ satisfies $F[P] \in \mathbb{R}$ and $D^c(P\|P) < \epsilon$. Therefore Slater's constraint qualification condition holds (see, e.g., Theorem 3.11.2 in Ponstein (2004)) and we can conclude strong duality

$$\sup_{Q : D^c(Q\|P) \leq \epsilon} \{E_Q[\mathcal{L}] - \kappa D^c(Q\|P)\} \tag{67}$$
$$= \inf_{\lambda > 0} \{\lambda \epsilon + \sup_{Q : D^c(Q\|P) < \infty} \{E_Q[\mathcal{L}] - (\kappa + \lambda) D^c(Q\|P)\}\} .$$

We note that the infimum can be restricted to $\lambda > 0$ (rather than $\lambda \geq 0$) due to the lower bound on the constraint function, $D^c(\cdot \| P) \geq 0$. This proves the claim. $\qquad \square$

If $D$ is an $f$-divergence, the convex conjugate term (i.e., the supremum over $Q$) in (65) can be evaluated by the same method as in Corollary A.20 and the result is a two-dimensional convex optimization problem.

**Corollary A.23.** *Suppose we have the following:*

1. *A measurable function $\mathcal{L} : \mathcal{X} \to [-\infty, \infty]$ that is bounded below or is bounded above.*

2. *A distribution $P \in \mathcal{P}(\mathcal{X})$ such that $\mathcal{L}^- \in L^1(P)$.*

3. *$f \in \mathcal{F}_1(a, b)$ where $a \geq 0$.*

4. *A cost function, $c$, that satisfies $c(x, x) = 0$ for all $x \in \mathcal{X}$.*

*Define $f^*(\pm\infty) := \infty$. Then for $\epsilon > 0$, $\kappa \geq 0$ we have*

$$\sup_{Q : D^c_f(Q\|P) \leq \epsilon} \{E_Q[\mathcal{L}] - \kappa D^c_f(Q\|P)\} \tag{68}$$
$$= \inf_{\lambda > 0, \rho \in \mathbb{R}} \{\lambda \epsilon + \rho + (\lambda + \kappa) E_P[f^*(\mathcal{L}^c_{\lambda+\kappa} - \rho/(\lambda + \kappa))]\}$$

*and the objective function for the minimization, $(0, \infty) \times \mathbb{R} \to (-\infty, \infty]$,*

$$(\lambda, \rho) \mapsto \lambda \epsilon + \rho + (\lambda + \kappa) E_P[f^*(\mathcal{L}^c_{\lambda+\kappa} - \rho/(\lambda + \kappa))] , \tag{69}$$

*is convex.*

*Proof.* Equation (68) follows from applying Theorem A.22 to $D = D_f$ and then evaluating the convex conjugate of $D_f(\cdot\|P)$ by the same method as in Corollary A.20. To prove convexity of the objective function, first note that for all $x$ the maps $h_x(t) := \sup_y\{t\mathcal{L}(y) - c(x,y)\}$ are convex in $t \in (0,\infty)$ and either $h_x > -\infty$ or $h_x(t) = -\infty$ for all $t$. $f^*$ is convex and it is straightforward to check that $a \geq 0$ implies $f^*$ is non-decreasing on $(-\infty,\infty]$. These facts together imply that $(t,\rho) \mapsto f^*(h_x(t) - \rho)$ are convex on $(0,\infty) \times \mathbb{R}$ for all $x$. Linearity of the expectation then implies $H(t,\rho) := E_P[f^*(h_x(t) - \rho)]$ is convex (note that $f^*(z) \geq z$ and so the assumptions on $\mathcal{L}$ imply that $H(t,\rho) > -\infty$). Therefore the perspective of $H$, given by $(\lambda,t,\rho) \mapsto \lambda H(t/\lambda, \rho/\lambda)$, is convex on $(0,\infty) \times (0,\infty) \times \mathbb{R}$. Composing with the affine map $(\lambda,\rho) \mapsto (\lambda + \kappa, 1, \rho)$ and adding the linear term $\lambda\epsilon + \rho$ results in a convex function on $(0,\infty) \times \mathbb{R}$. Substituting in the definitions of $H$ and $h_x$ we see that this function equals (69), thereby completing the proof. $\square$

Finally, we derive limiting formulas for the DRO problem, analogous to the interpolation results for $D^c$, Theorems A.12 and A.13. Though we don't use those theorems directly, the method of proof is similar and the conclusions align with what one expects in light of those results. We do require the more stringent assumptions that $\mathcal{X}$ is compact and $\mathcal{L}$ is upper semicontinuous (USC), which are often the case in practice.

**Theorem A.24.** *Suppose the Polish space $\mathcal{X}$ is compact and that we have the following:*

1. *An USC function $\mathcal{L} : \mathcal{X} \to [-\infty,\infty)$.*

2. *A distribution $P \in \mathcal{P}(\mathcal{X})$.*

3. *A pre-divergence $D$ such that $D(\cdot\|P)$ is LSC and $D(\mu_n\|P) \to 0$ implies $\mu_n \to P$ weakly.*

4. *A cost-function $c$.*

*For $r > 0$ define the cost functions $c_r = rc$. Then for $\epsilon > 0$ we have*

$$\lim_{r\to 0^+} \sup_{Q:D^{c_r}(Q\|P)\leq r\epsilon} E_Q[\mathcal{L}] = \sup_{Q:C(P,Q)\leq\epsilon} E_Q[\mathcal{L}]. \tag{70}$$

*If $\mathcal{L}_\theta : \mathcal{X} \to [-\infty,\infty)$, $\theta \in \Theta$, is a family of USC functions then*

$$\lim_{r\to 0^+} \inf_{\theta\in\Theta} \sup_{Q:D^{c_r}(Q\|P)\leq r\epsilon} E_Q[\mathcal{L}_\theta] = \inf_{\theta\in\Theta} \sup_{Q:C(P,Q)\leq\epsilon} E_Q[\mathcal{L}_\theta]. \tag{71}$$

*Proof.* Compactness of $\mathcal{X}$ and upper semicontinuity of $\mathcal{L}$ implies that $\mathcal{L}$ has a maximizer and hence $\mathcal{L}$ is bounded above. In particular, the expectations in (70) are all well-defined. The bound $D^{c_r}(Q\|P) \leq rC(P,Q)$ implies

$$\sup_{Q:C(P,Q)\leq\epsilon} E_Q[\mathcal{L}] \leq \sup_{Q:D^{c_r}(Q\|P)\leq r\epsilon} E_Q[\mathcal{L}] \tag{72}$$

for all $r > 0$. Define $K_r := \{Q : D^{c_r}(Q\|P) \leq r\epsilon\}$ and note that $r_2 \leq r_1$ implies $K_{r_2} \subset K_{r_1}$, hence the right-hand side of (72) is non-decreasing in $r$. $D(\cdot\|P)$ is LSC, therefore it has closed sublevel sets. $\mathcal{X}$ is a compact Polish space, therefore $\mathcal{P}(\mathcal{X})$ is compact (see, e.g., page 117 of Bogachev (2018)). Therefore the sublevel sets of $D(\cdot\|P)$ are also compact. Theorem A.11 then implies $D^{c_r}(\cdot\|P)$ is LSC for all $r$ and so the $K_r$ are closed sets, hence also compact. $\mathcal{L}$ is USC and bounded above, therefore the Portmanteau theorem implies that $Q \mapsto E_Q[\mathcal{L}]$ is USC, hence it achieves its maximum on $K_r$, i.e., there exists $Q_r \in K_r$ such that

$$\sup_{Q\in K_r} E_Q[\mathcal{L}] = E_{Q_r}[\mathcal{L}]. \tag{73}$$

Take $r_n \searrow 0^+$. Compactness of $\mathcal{P}(\mathcal{X})$ implies the existence of a weakly convergent subsequence $Q_j := Q_{r_{n_j}} \to Q_*$. Upper semicontinuity then implies $\limsup_j E_{Q_j}[\mathcal{L}] \leq E_{Q_*}[\mathcal{L}]$. By Theorem A.9 there exist $\eta_{*,j} \in \mathcal{P}(\mathcal{X})$ such that

$$\epsilon \geq \frac{1}{r_{n_j}} D^{c_{r_{n_j}}}(Q_j\|P) = \frac{1}{r_{n_j}} D(\eta_{*,j}\|P) + C(\eta_{*,j}, Q_j). \tag{74}$$

In particular we see that $\lim_{j \to \infty} D(\eta_{*,j} \| P) = 0$. The assumptions on $D$ therefore imply that $\eta_{*,j} \to P$ weakly. Now we can use lower semicontinuity of $C$ to compute

$$C(P, Q_*) \leq \liminf_j C(\eta_{*,j}, Q_j) \leq \liminf_j \frac{1}{r_{n_j}} D^{c_{r_{n_j}}}(Q_j \| P) \leq \epsilon. \tag{75}$$

Therefore $Q^* \in \{Q : C(P, Q) \leq \epsilon\}$. Putting these pieces together we find

$$\sup_{Q:C(P,Q)\leq\epsilon} E_Q[\mathcal{L}] \geq E_{Q^*}[\mathcal{L}] \geq \limsup_j E_{Q_{r_{n_j}}}[\mathcal{L}] = \limsup_j \sup_{Q \in K_{r_{n_j}}} E_Q[\mathcal{L}] \tag{76}$$

$$\geq \inf_{r>0} \sup_{Q:D^{c_r}(Q\|P)\leq r\epsilon} E_Q[\mathcal{L}] = \lim_{r\to 0^+} \sup_{Q:D^{c_r}(Q\|P)\leq r\epsilon} E_Q[\mathcal{L}],$$

where the last equality follows from the fact that the right-hand side is non-decreasing in $r$. Combining this inequality with (72) completes the proof of (70). If one has a family of USC functions $\mathcal{L}_\theta$ then apply (70) for each $\theta$ and note that the limit as $r \to 0^+$ is an infimum, hence one can commute the infimum over $\theta$ with the limit $r \to 0^+$ to obtain (71). □

**Theorem A.25.** *Suppose the Polish space $\mathcal{X}$ is compact and that we have the following:*

1. *An USC function $\mathcal{L} : \mathcal{X} \to [-\infty, \infty)$.*

2. *A distribution $P \in \mathcal{P}(\mathcal{X})$.*

3. *A pre-divergence $D$ such that $D(\cdot\|P)$ is LSC.*

4. *A cost-function $c$ that satisfies $c(x_1, x_2) = 0$ iff $x_1 = x_2$.*

*For $r > 0$ define the cost functions $c_r = rc$. Then for $\epsilon > 0$ we have*

$$\lim_{r\to\infty} \sup_{Q:D^{c_r}(Q\|P)\leq\epsilon} E_Q[\mathcal{L}] = \sup_{Q:D(Q\|P)\leq\epsilon} E_Q[\mathcal{L}]. \tag{77}$$

*If $\mathcal{L}_\theta : \mathcal{X} \to [-\infty, \infty)$, $\theta \in \Theta$, is a family of USC functions then*

$$\lim_{r\to\infty} \inf_{\theta\in\Theta} \sup_{Q:D^{c_r}(Q\|P)\leq\epsilon} E_Q[\mathcal{L}_\theta] = \inf_{\theta\in\Theta} \sup_{Q:D(Q\|P)\leq\epsilon} E_Q[\mathcal{L}_\theta]. \tag{78}$$

The proof of Theorem A.25 is similar to that of Theorem A.24, with slight differences that are motivated by the proof of Theorem A.13; we omit the details.

## A.3 WEAK CONVERGENCE AND $f$-DIVERGENCES

In this section we show that $f$-divergences can be used to prove weak convergence of measures; this is needed in Theorem A.7 as well as to apply many of the properties from Appendix A.1 to OT-regularized $f$-divergences. In fact, we will prove the stronger setwise convergence property. In this section we let $\mathcal{M}_b(\Omega)$ denote the set of bounded measurable real-valued functions on a measurable space $(\Omega, \mathcal{M})$.

**Definition A.26.** *Let $\{\mu_n\}_{n=1}^\infty, \mu$ be probability measures on the measurable space $(\Omega, \mathcal{M})$. We say that $\mu_n \to \mu$ **setwise** if $\lim_{n\to\infty} \mu_n(A) = \mu(A)$ for all $A \in \mathcal{M}$.*

First recall that setwise convergence implies convergence of integrals; we provide a simple proof of this fact.

**Lemma A.27.** *Let $(\Omega, \mathcal{M})$ be a measurable space and $\mu_n$, $\mu$ be probability measures on $\Omega$. If $\mu_n \to \mu$ setwise then $\int \phi d\mu_n \to \int \phi d\mu$ for all $\phi \in \mathcal{M}_b(\Omega)$.*

**Remark A.28.** *In particular, if $(\Omega, \mathcal{M})$ is a metric space with the Borel $\sigma$-algebra then this implies $\mu_n \to \mu$ weakly.*

*Proof.* Let $\phi \in \mathcal{M}_b(\Omega)$. Take a sequence of simple functions $\phi_j$ that converge uniformly to $\phi$ (see, e.g., Theorem 2.10 in Folland (2013)). With these we can compute

$$\left| \int \phi d\mu_n - \int \phi d\mu \right| \leq \left| \int \phi d\mu_n - \int \phi_j d\mu_n \right| + \left| \int \phi_j d\mu_n - \int \phi_j d\mu \right| + \left| \int \phi_j d\mu - \int \phi d\mu \right|$$

$$\leq \|\phi - \phi_j\|_\infty (\mu_n(\Omega) + \mu(\Omega)) + \left| \int \phi_j d\mu_n - \int \phi_j d\mu \right|. \tag{79}$$

The fact that $\phi_j$ are simple and $\mu_n \to \mu$ setwise implies that $|\int \phi_j d\mu_n - \int \phi_j d\mu| \to 0$ as $n \to \infty$ for all $j$, hence

$$\limsup_{n \to \infty} |\int \phi d\mu_n - \int \phi d\mu| \leq 2\|\phi - \phi_j\|_\infty \tag{80}$$

for all $j$. Taking $j \to \infty$ completes the proof. $\qquad\square$

We now prove that convergence of an $f$-divergence to zero implies setwise convergence under mild assumptions on $f$.

**Theorem A.29.** *Let $(\Omega, \mathcal{M})$ be a measurable space, $f \in \mathcal{F}_1(a, b)$, and define $w_0 := f'_+(1)$ (where $f'_+$ denotes the right derivative of $f$, which exists because $f$ is convex). Suppose $w_0 \in \{f^* < \infty\}^o$ ($A^o$ denotes the interior of the set $A$) and $f$ is strictly convex on a neighborhood of $1$. If $P_n, P$ are probability measures on $\Omega$ and either $D_f(P_n\|P) \to 0$ or $D_f(P\|P_n) \to 0$ then $P_n \to P$ setwise.*

*If $(\Omega, \mathcal{M})$ is a metric space with the Borel $\sigma$-algebra then we can further conclude $P_n \to P$ weakly.*

*Proof.* Take any probability measures $Q_1, Q_2$ on $(\Omega, \mathcal{M})$ and $A \in \mathcal{M}$. For all $\epsilon > 0$ we define $\phi_\epsilon = w_0 + \epsilon 1_A$. Then $\phi_\epsilon \in \mathcal{M}_b(\Omega)$, hence the variational representation of $f$-divergences (see Proposition B.1 in Birrell et al. (2022)) implies

$$\begin{aligned} D_f(Q_1\|Q_2) &\geq E_{Q_1}[\phi_\epsilon] - E_{Q_2}[f^*(\phi_\epsilon)] \\ &= w_0 + \epsilon Q_1(A) - E_{Q_2}[f^*(w_0 + \epsilon 1_A)]. \end{aligned} \tag{81}$$

We have assumed that $w_0 \in \{f^* < \infty\}^o$, hence there exists $\delta > 0$ with $B_\delta(w_0) \subset \{f^* < \infty\}$. Using properties of the Taylor expansion of convex functions (see Liese & Vajda (2006)) along with the identities $f^*(w_0) = w_0$ and $(f^*)'_+(w_0) = 1$ (see Lemma A.9 in Birrell et al. (2022)) we can compute

$$\begin{aligned} f^*(y) &= f^*(w_0) + (f^*)'_+(w_0)(y - w_0) + R_{f^*}(w_0, y) \\ &= y + R_{f^*}(w_0, y) \\ &\leq y + |y - w_0||(f^*)'_+(y) - (f^*)'_+(w_0)| \end{aligned} \tag{82}$$

for all $y \in B_\delta(w_0)$. Letting $\epsilon < \delta$ we have range$(w_0 + \epsilon 1_A) \subset B_\delta(w_0)$ and so

$$\begin{aligned} f^*(w_0 + \epsilon 1_A) &\leq w_0 + \epsilon 1_A + |w_0 + \epsilon 1_A - w_0||(f^*)'_+(w_0 + \epsilon 1_A) - (f^*)'_+(w_0)| \\ &= w_0 + \epsilon 1_A + \epsilon 1_A |(f^*)'_+(w_0 + \epsilon) - (f^*)'_+(w_0)|. \end{aligned} \tag{83}$$

Hence

$$\begin{aligned} D_f(Q_1\|Q_2) &\geq w_0 + \epsilon Q_1(A) - E_{Q_2}[w_0 + \epsilon 1_A + \epsilon 1_A |(f^*)'_+(w_0 + \epsilon) - (f^*)'_+(w_0)|] \\ &= \epsilon[Q_1(A) - Q_2(A)(1 + |(f^*)'_+(w_0 + \epsilon) - (f^*)'_+(w_0)|)]. \end{aligned} \tag{84}$$

Now let $P_n, P$ be probability measures on $\Omega$ and consider the following two cases.

1. Suppose $D_f(P_n, P) \to 0$. Then letting $Q_1 = P_n$ and $Q_2 = P$ in the above we get

$$\begin{aligned} 0 &= \limsup_n D_f(P_n\|P) \tag{85} \\ &\geq \epsilon[\limsup_n P_n(A) - P(A)(1 + |(f^*)'_+(w_0 + \epsilon) - (f^*)'_+(w_0)|)] \tag{86} \end{aligned}$$

    for all $\epsilon \in (0, \delta)$. If $\limsup_n P_n(A) > P(A)$ then by right-continuity of $(f^*)'_+$ (the right-derivative of a convex function), for $\epsilon$ small enough the term in brackets in (86) is positive, which is a contradiction. Therefore $\limsup_n P_n(A) \leq P(A)$. This holds for all $A \in \mathcal{M}$, hence for a given $A$ we can apply it to $A^c$ to get $\limsup_n P_n(A^c) \leq P(A^c)$, hence $\liminf_n P_n(A) \geq P(A)$. Together these bounds imply $\lim_n P_n(A) = P(A)$ for all $A \in \mathcal{M}$, hence $P_n \to P$ setwise.

2. Suppose $D_f(P, P_n) \to 0$. Letting $Q_1 = P$ and $Q_2 = P_n$ we have

$$\begin{aligned} 0 &= \limsup_n D_f(P\|P_n) \tag{87} \\ &\geq \epsilon[P(A) - \liminf_n P_n(A)(1 + |(f^*)'_+(w_0 + \epsilon) - (f^*)'_+(w_0)|)]. \end{aligned}$$

If $P(A) > \liminf_n P_n(A)$ then for $\epsilon$ sufficiently small we again find the term in brackets to be positive, which is a contradiction. Hence $P(A) \leq \liminf_n P_n(A)$ for all $A \in \mathcal{M}$. Applying this to $A^c$ and combining the results gives $\lim_{n\to\infty} P_n(A) = P(A)$ for all $A \in \mathcal{M}$. Hence $P_n \to P$ setwise.

If $(\Omega, \mathcal{M})$ is a metric space with the Borel $\sigma$-algebra then we can further conclude $P_n \to P$ weakly by using Lemma A.27. $\qquad\square$

## B INTERPRETING THE OUTER MINIMIZER: ADVERSARIAL SAMPLE WEIGHTS

In this appendix we derive the (formal) solution (16) to the optimization problem (14) that was presented in Section 2.1 above. We work under the assumptions that $\mathcal{X} = \mathbb{R}^d$, exact optimizers exist, and all functions are sufficiently smooth.

Begin by letting $y_i(\lambda)$ be the solution to the inner maximizer (12) with $x = x_i$ as a function of $\lambda$ and let $\lambda_*$ and $\rho_*$ be the optimal scaling and shift parameters for the outer minimizer at a fixed $\theta$ (we suppress the $\theta$-dependence of $y_i$, $\lambda_*$, and $\rho_*$ in the notation). Taking the gradient of the objective function for the inner maximizer (12) with respect to $y$ and evaluating at the optimizer $y_i(\lambda)$, we find

$$\lambda^{-1}\nabla_y \mathcal{L}_\theta(y_i(\lambda)) - \nabla_y c(x_i, y_i(\lambda)) = 0 \tag{88}$$

for all $\lambda$. Differentiating the objective function in (95) with respect to $\rho$ we find

$$\partial_\rho|_{\rho=\rho_*}\left(\epsilon\lambda_* + \rho + \lambda_*\frac{1}{n}\sum_{i=1}^n f^*(\mathcal{L}^c_{\theta,\lambda_*}(x_i) - \rho/\lambda_*)\right) \tag{89}$$

$$= 1 + \lambda_*\frac{1}{n}\sum_{i=1}^n (f^*)'(\mathcal{L}^c_{\theta,\lambda_*}(x_i) - \rho_*/\lambda_*)(-\lambda_*^{-1}) = 0\,,$$

i.e.,

$$\frac{1}{n}\sum_{i=1}^n (f^*)'(\mathcal{L}^c_{\theta,\lambda_*}(x_i) - \rho_*/\lambda_*) = 1\,. \tag{90}$$

In particular, this implies that the $p_{*,i}$'s, defined by

$$p_{*,i} := \frac{1}{n}(f^*)'(\mathcal{L}^c_{\theta,\lambda_*}(x_i) - \rho_*/\lambda_*)\,, \tag{91}$$

sum to 1. We next differentiate the objective function with respect to $\lambda$ to obtain

$$\epsilon + \frac{1}{n}\sum_i f^*(\mathcal{L}^c_{\theta,\lambda_*}(x_i) - \rho_*/\lambda_*) \tag{92}$$

$$+ \lambda\frac{1}{n}\sum_i (f^*)'(\mathcal{L}^c_{\theta,\lambda_*}(x_i) - \rho_*/\lambda_*)(\partial_\lambda|_{\lambda=\lambda_*}\mathcal{L}^c_{\theta,\lambda}(x_i) + \rho_*/\lambda_*^2) = 0\,,$$

where we can use (88) to simplify

$$\partial_\lambda\mathcal{L}^c_{\theta,\lambda}(x_i) = -\lambda^{-2}\mathcal{L}_\theta(y_i(\lambda)) + (\lambda^{-1}\nabla_y\mathcal{L}_\theta(y_i(\lambda)) - \nabla_y c(x_i, y_i(\lambda))) \cdot y_i'(\lambda) \tag{93}$$

$$= -\lambda^{-2}\mathcal{L}_\theta(y_i(\lambda))\,.$$

Combining (90), (92), and (93) we can compute

$$\epsilon\lambda_* + \rho_* + \lambda_*\frac{1}{n}\sum_i f^*(\mathcal{L}^c_{\theta,\lambda_*}(x_i) - \rho_*/\lambda_*) \tag{94}$$

$$= \frac{1}{n}\sum_i (f^*)'(\mathcal{L}^c_{\theta,\lambda_*}(x_i) - \rho_*/\lambda_*)(\mathcal{L}_\theta(y_i(\lambda_*)) - \rho_*) - \lambda_*\frac{1}{n}\sum_i f^*(\mathcal{L}^c_{\theta,\lambda_*}(x_i) - \rho_*/\lambda_*)$$

$$+ \rho_* + \lambda_*\frac{1}{n}\sum_i f^*(\mathcal{L}^c_{\theta,\lambda_*}(x_i) - \rho_*/\lambda_*)$$

$$= \frac{1}{n}\sum_i (f^*)'(\mathcal{L}^c_{\theta,\lambda_*}(x_i) - \rho_*/\lambda_*)\mathcal{L}_\theta(y_i(\lambda_*))\,.$$

Recalling the definition of $\lambda_*$ and $\rho_*$, this implies

$$\inf_{\lambda>0,\rho\in\mathbb{R}}\left\{\epsilon\lambda+\rho+\lambda\frac{1}{n}\sum_{i=1}^{n}f^*(\mathcal{L}_{\theta,\lambda}^c(x_i)-\rho/\lambda)\right\}=E_{Q_{*,\theta}}[\mathcal{L}_\theta]\,,\tag{95}$$

where the optimal adversarial distribution is

$$Q_{*,\theta}:=\sum_{i=1}^{n}\frac{1}{n}(f^*)'(\mathcal{L}_{\theta,\lambda_*}^c(x_i)-\rho_*/\lambda_*)\delta_{y_i(\lambda_*)}\,.\tag{96}$$

This is the equality claimed above in (95). See Section 2.1 for a discussion of the implications of this formula.

## B.1 RELATION TO OT-BASED ADVERSARIAL TRAINING

The robust-optimization adversarial training method (1) is a special case of OT-DRO, as was noted previously in Regniez et al. (2021); Bui et al. (2022), with a specific choice of cost function. The $ARMOR_D$ methods, which allow for general OT cost, are therefore generalizations (1). In addition, the inclusion of an information-theoretic component in $ARMOR_D$ implies that it is also an extension of the general OT-DRO method Bui et al. (2022), allowing for distribution neighborhoods which have qualitatively different structure. This allows for qualitatively new types of robustness (i.e., beyond the shifting of samples within an allowed neighborhood), such as miss-specification of the mass of widely-separated modes. Algorithmically, this feature effectively introduces dynamical sample weights (91), as shown via the formal solution (96)-(96). In short, adversarial samples in the $ARMOR_D$ method are both transported (as in other OT-based methods) and re-weighted, with the latter being the new ingredient in our method. This new feature contributes non-trivially to the performance of the $ARMOR_D$ methods (see Appendix C.8).

## C IMPLEMENTATION DETAILS

To foster reproducability of our results, we provide the threat model, pseudocode for the OT-regularized-divergence adversarial robustness methods, the target network structures used in the malware and image applications, and the hyperparameters that yielded the results in Tables 3, 2, 4, and 5.

## C.1 THREAT MODEL

Following the guidelines in Carlini et al. (2019), we consider the following threat model characterizing the adversary's goal, knowledge, and capabilities in implementing $ARMOR_D$ as a method for enhancing adversarial robustness.

1. *Adversaries goal:* The adversaries goal is to generate adversarial samples that force the image/malware detector to make erroneous predictions. To avoid restrictive assumptions, any wrong classification is considered as a successful attack.

2. *Adversary's knowledge:* To avoid restrictive assumptions, we assume that the adversary has complete knowledge of the inner workings of the target model (i.e., white-box access). This aligns with the Kerckhoff's principle that mandates security even if system details are known to the adversary.

3. *Adversary's Capabilities:* The adversary can apply arbitrary modifications to natural samples of any class (e.g., both malicious and benign in the binary classification case).

## C.2 ALGORITHM PSEUDOCODE

In this appendix we provide pseudocode for the methods proposed in this work. We will refer to the objective functions (97)- (100) therein, where $\phi_\theta$ denotes the target model (i.e., classifier) and $CE$ denotes the cross-entropy loss. The following applies to the adversarial samples methods (i.e., $adv_s$), obtained from the cost function (22); the generalization to both adversarial samples and labels (i.e., $adv_{s,l}$) using (24) is described following Algorithm 1.

1. **Inner Maximizer Objective with Adversarial Samples (see Eq. 23):**

$$A_s(x, \tilde{x}, y, \lambda, \theta) := \lambda^{-1} CE(\phi_\theta(\tilde{x}), y) - L\|x - \tilde{x}\|^q. \tag{97}$$

2. **Inner Maximizer Objective with Adversarial Labels and Samples (see Eq. 25):**

$$A_{s,l}(x, \tilde{x}, y, \tilde{y}, \lambda, \theta) := \lambda^{-1}(CE(\phi_\theta(\tilde{x}), \tilde{p}) - CE(\tilde{y}, \tilde{p})) - L\|x - \tilde{x}\|^q - Kg_\delta(1 - \tilde{p}_k), \tag{98}$$

where $g_\delta(z) = z/(1 - z/\delta)$, $\tilde{p} = Softmax(\tilde{y})$ is the adversarial label probability-vector, $y = 1_k$ is a one-hot encoded label corresponding to the class being $k$, and $\tilde{p}_k$ is the $k$'th component of $\tilde{p}$.

3. **KL-Divergence Outer Minimizer Objective (see Eq. 15):**

$$A_w^{KL}(x, \tilde{x}, y, \lambda, \theta) := \epsilon\lambda + \lambda \log\left(\frac{1}{B}\sum_{i=1}^{B}\exp(A_s(x_i, \tilde{x}_i, y_i, \lambda, \theta))\right). \tag{99}$$

4. **$f$-Divergence Outer Minimizer Objective (see Eq. 14):**

$$A_w^f(x, \tilde{x}, y, \lambda, \rho, \theta) := \epsilon\lambda + \rho + \lambda\frac{1}{B}\sum_{i=1}^{B}f^*(A_s(x_i, \tilde{x}_i, y_i, \lambda, \theta) - \rho/\lambda). \tag{100}$$

In the examples in Section 3 we use the $\alpha$-divergences, for which $f_\alpha^*$ is given in (30).

---

**Algorithm 1** **A**dversarially **R**obust Deep Learning **M**odels with **O**ptimal-Transport-**R**egularized **D**ivergences ($ARMOR_D$)

---

**Input:** Labeled training data $\{x_i, y_i\}$, target model $\phi_\theta$ depending on NN-parameters $\theta$, number of training epochs $N$, minibatch size $B$, information divergence $D$ ($D = KL$ or $D = D_f$), number of inner maximizer iterations $M$, learning rates $lr_{\tilde{x}}$, $lr_\lambda$, and $lr_\theta$, other hyperparameters listed in Appendix C.5.
**Output:** robustified model $\phi_\theta$
  1: **for** $n = 1, \ldots, N$ **do**
  2:     Sample a minibatch $B_n$
  3:     **for** $(x_i, y_i) \in B_n$ **do**
  4:         $\tilde{x}_i \leftarrow x_i + $noise
  5:         **for** $m = 1, \ldots, M$ **do**
  6:             $\tilde{x}_i \leftarrow \tilde{x}_i + lr_{\tilde{x}}\nabla_{\tilde{x}}A_s(x_i, \tilde{x}_i, y_i, \lambda, \theta)$                                  ▷ See (97)
  7:         **end for**
  8:     **end for**
  9:     **if** $D = KL$ **then**
 10:         $\lambda \leftarrow \lambda - lr_\lambda\nabla_\lambda A_w^{KL}(x, \tilde{x}, y, \lambda, \theta)$                               ▷ See (99)
 11:         $\theta \leftarrow \theta - lr_\theta\nabla_\theta A_w^{KL}(x, \tilde{x}, y, \lambda, \theta)$
 12:     **else if** $D = D_f$ **then**
 13:         $(\lambda, \rho) \leftarrow (\lambda, \rho) - lr_\lambda\nabla_{(\lambda,\rho)}A_w^f(x, \tilde{x}, y, \lambda, \rho, \theta)$               ▷ See (100)
 14:         $\theta \leftarrow \theta - lr_\theta\nabla_\theta A_w^f(x, \tilde{x}, y, \lambda, \rho, \theta)$
 15:     **end if**
 16: **end for**

---

Lines 3-8 of Algorithm 1 implement the inner maximizer, wherein the adversarial samples $\tilde{x}_i$ are constructed and lines 9-15 implement one step of the outer minimizer. In line 4 of the inner maximizer we allow for noise to be added to the natural sample when initializing the adversarial sample; default is IID Uniform$[-lr_{\tilde{x}}, lr_{\tilde{x}}]$ noise added to each component. To incorporate adversarial labels into Algorithm 1 one can make the following modifications. First, after line 4 in Algorithm 1 add the initialization

$$\tilde{y}_i \leftarrow \log((N_c - 1)(2 - \delta)/\delta)y_i, \tag{101}$$

where $N_c$ is the total number of classes and $y_i$ is the one-hot encoded label for the sample $x_i$; this corresponds to initializing the adversarial label probabilities so that the probability of the given

class is $1 - \delta/2$, with the remaining probability-mass equally distributed over the other classes (other initialization strategies are certainly possible, but we did not experiment with any alternatives). Second, within the inner maximizer replace $A_s$ in line 6 with $A_{s,l}$ (98) and after line 6 add the update

$$\tilde{y}_i \leftarrow \tilde{y}_i + lr_{\tilde{y}} \nabla_{\tilde{y}} A_{s,l}(x_i, \tilde{x}_i, y_i, \tilde{y}_i, \lambda, \theta). \tag{102}$$

Finally, in the outer minimizers, replace $A_s$ with $A_{s,l}$ (98) in the definitions of $A_w^{KL}$ (99) in lines 10-11 and $A_w^f$ (100) in lines 13-14.

The modifications to Algorithm 1 (specifically, to the outer minimizers) that yield the $adv + nat$ and $adv^a$ methods are outlined in Appendices C.6 and C.7 respectively below.

**Fail-safe Mechanisms Used in Training** In our implementation we adopted the following two fail-safe mechanisms to ensure that quantities remain in their specified domains:

- In the rare instance that $\lambda$ becomes negative due to large learning rates, we used a fail-safe mechanism in which we applied the Softplus function, $Softplus(z) = \log(1 + exp(z))$ to $\lambda$ to ensure it remains positive during the learning process.
- $adv_{s,l}$ methods: In the rare event that in (25) we obtain $\tilde{p}_k \leq 1 - \delta$ (where $y = 1_k$ is the one-hot encoded label) due to large learning rates in updating $\tilde{y}$ (where $\tilde{p} = Softmax(\tilde{y})$) we override the updated value by resetting it to the same value used to initialize the adversarial training loop; see (101).

### C.3    TARGET NETWORKS' STRUCTURE

The malware detector is a feed-forward neural network with 3 hidden layers each with 300 ReLU-activated neurons. The output layer uses a negative log-likelihood loss and training of the NN-parameters was done using the ADAM optimization with a 0.001 learning rate, a minibatch of size 16, learned over 150 epochs Al-Dujaili et al. (2018). For the MNIST dataset, we followed Bui et al. (2022) with the network structure from Carlini & Wagner (2017). The MNIST image detector is a CNN network with four convolutional layers (32, 32, 64, and 64) each with a square filter of size 3 and ReLU activations, two $2 \times 2$ max-pooling layers, and three fully connected layers with a dropout layer between the first and second fully connected layer (Carlini & Wagner (2017)). This network was trained with SGD optimizer over 100 epochs with a starting learning rate of $1e - 2$ reduced by $\times 0.1$ at epochs $\{55, 70, 90\}$.

### C.4    PARAMETER SEARCH SPACE AND SELECTED HYPERPARAMETERS FOR SECTION 3.1

1. The distribution neighborhood size $\epsilon > 0$ The value was selected from $\epsilon \in \{2e - 1, 2.2e - 1, 2.4e - 1, 2.5e - 1, 3e - 1\}$.

2. The $\alpha$-divergence parameter (29) The value was selected from $\alpha \in \{2, 2.5, 3, 3.5, 7\}$.

3. The learning rate $lr_\lambda$ used for both $\lambda$ and $\rho$, the real parameters in the outer minimizers (14) and (15) The value was selected from $lr_\lambda \in \{8e - 4, 1.5e - 3, 1.6e - 3, 2e - 3\}$.

   **Hyperparameters used by $ARMOR_\alpha$-$PGD$ in Table 1**
   $ARMOR_\alpha$ ($adv_s + nat$): $\epsilon = 2.4e - 1$, $\alpha = 3$, and $lr_\lambda = 1.6e - 3$.

### C.5    PARAMETER SEARCH SPACE AND SELECTED HYPERPARAMETERS FOR SECTION 3.2 AND APPENDIX D

1. The distribution neighborhood size $\epsilon > 0$ in (14)-(15):
   The value was selected from $\epsilon \in \{1e - 4, 2e - 4, 4e - 4, 5e - 4, 6e - 4, 7e - 4, 8e - 4, 1e - 3, 2e - 3, 4e - 3, 5e - 3, 6e - 3, 7e - 3, 8e - 3, 1e - 2, 2e - 2, 4e - 2, 5e - 2, 6e - 2, 7e - 2, 8e - 2, 1e - 1, 2e - 1, 3e - 1, 4e - 1, 5e - 1, 6e - 1, 7e - 1, 8e - 1, 1e - 2, 1e - 1, 1.0, 2.0, 2.5, 3.0, 3.5, 4.0, 4.5, 5.0, 10.0\}$.

2. The coefficient $L > 0$ in the cost functions (22) and (24):
   The value was selected from $L \in \{1e - 5, 2e - 5, 5e - 5, 8e - 5, 1e - 4, 2e - 4, 5e - 4, 8e - 4, 1e - 3, 2e - 3, 5e - 3, 8e - 3, 1e - 2, 2e - 2, 3e - 2, 5e - 2, 8e - 2, 1e - 1, 2e - 1, 5e - 1, 8e - 1, 1.0, 2.0, 5.0, 10.0\}$.

3. The coefficient $K > 0$ in the cost function (24):
The value was selected from $K \in \{2e-3, 5e-3, 8e-3, 2e-2, 5e-2, 8e-2, 2e-1, 5e-1, 8e-1, 1.0, 1.5e1, 5e1, 8e1, 1e2, 1.5e2, 2e2, 4.5e2, 5e2, 1e3\}$

4. The parameter $\delta$ in the cost function (24):
The value was selected from $\delta \in \{1e-1, 2e-1, 3e-1, 4e-1\}$

5. The power $q > 0$ in the cost functions (22) and (24):
The value was selected from $q \in \{0.5, 1.0, 1.5, 2.0, 2.5, 3.0, 4.0, 5.0\}$.

6. The $\alpha$-divergence parameter (29), $\alpha > 1$:
The value was selected from $\alpha \in \{1.5, 1.8, 2.0, 2.5, 3.0, 3.5, 4.0, 5.0, 6.0\}$.

7. The vector norm in the cost functions (22) and (24) was selected from $\ell^1$, $\ell^2$, and $\ell^\infty$.

8. The learning rate $lr_\lambda$ used for both $\lambda$ and $\rho$, the real parameters in the outer minimizers (14) and (15):
The value was selected from $lr_\lambda \in \{2e-4, 5e-4, 8e-4, 2e-3, 5e-3, 8e-3, 2e-2, 5e-2, 8e-2, 1e-1, 2e-1\}$

9. The coefficient $t \in [0, 1]$ for the loss of original samples ($1 - t$ denotes the coefficient for the loss of adversarial samples) in the $nat$ methods (see Appendix C.6):
The value was selected from $t \in \{1e-1, 2e-1, \ldots, 9e-1\}$.

For the asymmetric methods (see Appendix C.7) we always use $s = 1/2$ and did not test other values. Following Kolter & Madry (2018), the step size parameter for learning $\tilde{x}$ in the inner maximizer for all MNIST experiments was fixed to 0.01. Following Al-Dujaili et al. (2018), this parameter was fixed to 0.02 for all malware experiments. In experiments with adversarial labels, i.e., $ARMOR_\alpha$ ($adv_{s,l}$) and $ARMOR_\alpha$ ($adv_{s,l} + nat$), the step size for learning $\tilde{y}$ was the same as for $\tilde{x}$ in all cases (i.e., 0.01 for MNIST and 0.02 for malware).

### C.5.1 Hyperparameters in Malware Experiments for Enhancing Adversarial Robustness: Table 2

- $ARMOR_{KL}$ ($adv_s$): $\epsilon = 1e-1$, $L = 1e-2$, $q = 2.0$, $lr_\lambda = 2e-3$, and $\ell^\infty$ norm.
- $ARMOR_{KL}$ ($adv_s^a$): $\epsilon = 1e-2$, $L = 5e-3$, $q = 1.5$, $lr_\lambda = 2e-3$, and $\ell^\infty$ norm.
- $ARMOR_{KL}$ ($adv_s + nat$): $\epsilon = 1e-1$, $L = 1e-2$, $q = 2.0$, $lr_\lambda = 2e-4$, $t = 0.5$, and $\ell^\infty$ norm.
- $ARMOR_{KL}$ ($adv_s^a + nat$): $\epsilon = 3.5$, $L = 1e-3$, $q = 2.0$, $lr_\lambda = 8e-3$, $t = 0.5$, and $\ell^\infty$ norm.
- $ARMOR_\alpha$ ($adv_s$): $\epsilon = 3.0$, $L = 3e-2$, $q = 2.0$, $\alpha = 2.5$, $lr_\lambda = 8e-4$, and $\ell^\infty$ norm.
- $ARMOR_\alpha$ ($adv_s^a$): $\epsilon = 1e-2$, $L = 1e-2$, $q = 2.0$, $\alpha = 1.5$, $lr_\lambda = 2e-4$, and $\ell^\infty$ norm.
- $ARMOR_\alpha$ ($adv_s + nat$): $\epsilon = 3.0$, $L = 3e-2$, $q = 1.5$, $lr_\lambda = 8e-4$, $\alpha = 2.5$, $t = 0.5$, and $\ell^\infty$ norm.
- $ARMOR_\alpha$ ($adv_s^a + nat$): $\epsilon = 3.0$, $L = 1e-3$, $q = 2.0$, $lr_\lambda = 8e-4$, $\alpha = 2.5$, $t = 0.5$, and $\ell^\infty$ norm.
- $ARMOR_\alpha$ ($adv_{s,l}$): $\epsilon = 3.0$, $L = 3e-2$, $q = 2.0$, $lr_\lambda = 8e-4$, $\alpha = 2.5$, $t = 0.5$, $\ell^\infty$ norm, $\delta = 0.4$, and $K = 450$.

### C.5.2 Hyperparameters in Image Experiments for Enhancing Adversarial Robustness: Table 3

- $ARMOR_{KL}$ ($adv_s$): $\epsilon = 5e-4$, $L = 1e-1$, $q = 1.5$, $lr_\lambda = 2e-4$, and $\ell^2$ norm.
- $ARMOR_{KL}$ ($adv_s + nat$): $\epsilon = 1.0$, $L = 8e-2$, $q = 2.0$, $lr_\lambda = 2e-3$, $t = 0.5$, and $\ell^\infty$ norm.
- $ARMOR_\alpha$ ($adv_s$): $\epsilon = 6e-4$, $L = 1e-1$, $q = 2.0$, $\alpha = 2.0$, $lr_\lambda = 5e-4$, and $\ell^2$ norm.
- $ARMOR_\alpha$ ($adv_s + nat$): $\epsilon = 2.0$, $L = 3e-2$, $q = 1.5$, $lr_\lambda = 8e-4$, $\alpha = 2.5$, $t = 0.5$, and $\ell^\infty$ norm.
- $ARMOR_\alpha$ ($adv_{s,l} + nat$): $\epsilon = 3.0$, $L = 3e-2$, $q = 1.5$, $lr_\lambda = 8e-4$, $\alpha = 2.5$, $t = 0.5$, and $\ell^2$ norm, $\delta = 0.1$, $K = 8e-2$.

### C.5.3 Hyperparameters in Image Experiments for Enhancing Test Generalizability: Table 4

- $ARMOR_{KL}$ ($adv_s$): $\epsilon = 5e-4$, $L = 1e-1$, $q = 1.5$, $lr_\lambda = 2e-4$, and $\ell^2$ norm.
- $ARMOR_{KL}$ ($adv_s + nat$): $\epsilon = 4e-3$, $L = 8e-3$, $q = 2.0$, $lr_\lambda = 2e-3$, $t = 0.5$, and $\ell^2$ norm.
- $ARMOR_\alpha$ ($adv_s$): $\epsilon = 5e-4$, $L = 1e-1$, $q = 2.5$, $\alpha = 5.0$, $lr_\lambda = 5e-4$, and $\ell^2$ norm.
- $ARMOR_\alpha$ ($adv_s + nat$): $\epsilon = 2.0$, $L = 3e-2$, $q = 1.5$, $lr_\lambda = 8e-4$, $\alpha = 2.5$, $t = 0.5$, and $\ell^\infty$ norm.
- $ARMOR_\alpha$ ($adv_{s,l} + nat$): $\epsilon = 3.0$, $L = 3e-2$, $q = 1.5$, $lr_\lambda = 8e-4$, $\alpha = 2.5$, $t = 0.5$, $\ell^\infty$ norm, $\delta = 0.1$, and $K = 8e-2$.

### C.5.4 Hyperparameters in Malware Experiments for Enhancing Test Generalizability: Table 5

- $ARMOR_{KL}$ ($adv_s$): $\epsilon = 7e-1$, $L = 5e-5$, $q = 2.0$, $lr_\lambda = 2e-3$, and $\ell^1$ norm.
- $ARMOR_{KL}$ ($adv_s^a$): $\epsilon = 1e-2$, $L = 5e-3$, $q = 1.5$, $lr_\lambda = 2e-3$, and $\ell^1$ norm.
- $ARMOR_{KL}$ ($adv_s + nat$): $\epsilon = 2e-1$, $L = 5e-5$, $q = 2.0$, $lr_\lambda = 2e-4$, $t = 0.5$, and $\ell^1$ norm.
- $ARMOR_{KL}$ ($adv_s^a + nat$): $\epsilon = 3.5$, $L = 1e-3$, $q = 2.0$, $lr_\lambda = 8e-3$, $t = 0.5$, and $\ell^\infty$ norm.
- $ARMOR_\alpha$ ($adv_s$): $\epsilon = 3.0$, $L = 3e-2$, $q = 2.0$, $\alpha = 2.5$, $lr_\lambda = 8e-4$, and $\ell^\infty$ norm.
- $ARMOR_\alpha$ ($adv_s^a$): $\epsilon = 2e-1$, $L = 5e-5$, $q = 2.0$, $\alpha = 2.5$, $lr_\lambda = 2e-4$, and $\ell^1$ norm.
- $ARMOR_\alpha$ ($adv_s + nat$): $\epsilon = 7e-1$, $L = 1e-2$, $q = 2.0$, $lr_\lambda = 2e-4$, $\alpha = 2.5$, $t = 0.5$, and $\ell^\infty$ norm.
- $ARMOR_\alpha$ ($adv_s^a + nat$): $\epsilon = 4e-1$, $L = 5e-5$, $q = 2.0$, $lr_\lambda = 2e-4$, $\alpha = 2.5$, $t = 0.5$, and $\ell^1$ norm.
- $ARMOR_\alpha$ ($adv_{s,l}$): $\epsilon = 4e-1$, $L = 5e-5$, $q = 2.0$, $lr_\lambda = 2e-4$, $\alpha = 2.5$, $t = 0.5$, $\ell^1$ norm, $\delta = 0.4$, and $K = 150$.

### C.6 Robust Optimization Using a Mixture of Adversarial and Natural Samples

The best performance in the experiments presented in Section 3 was often obtained using a mixture of adversarial samples along with the original training data (called the natural samples) and their corresponding losses. This can be viewed as DRO over distribution neighborhoods of the form

$$\mathcal{U}_{\epsilon,t}^{D^c}(P_n) := \{tP_n + (1-t)Q : D^c(Q\|P_n) \le \epsilon\}, \quad \epsilon > 0, t \in (0,1), \tag{103}$$

as we have

$$\inf_{\theta \in \Theta} \sup_{Q \in \mathcal{U}_{\epsilon,t}^{D^c}(P_n)} E_Q[\mathcal{L}_\theta] = \inf_{\theta \in \Theta} \left\{ tE_{P_n}[\mathcal{L}_\theta] + (1-t) \sup_{Q:D^c(Q\|P_n) \le \epsilon} E_Q[\mathcal{L}_\theta] \right\}. \tag{104}$$

The supremum over $Q$ on the right-hand side of (104) can then be evaluated by the method discussed in Section 2 and the resulting expression is used in what we call the $adv + nat$ methods. More specifically, the $adv_s$ variants refer to the use of adversarial samples constructed via the OT-regularized loss (23) while $adv_{s,l}$ refers to the use of adversarial samples together with adversarial labels, both of which are constructed via the OT-regularized loss (25). We generally find the best performance occurs when combining the natural loss, which incorporates all samples in each minibatch, with our method's adversarial re-weighting, which focuses on the most difficult samples.

### C.7 Asymmetric Robust Optimization

In many cases the training samples are naturally partitioned into distinct components, with corresponding empirical distributions $P_{n,0}$ and $P_{n,1}$ (e.g., distinct class labels), and one wishes to robustify only one component of the partition (e.g., to protect against false negative adversarial attacks but

not false positives). In such cases one can formulate the DRO problem in an asymmetric manner as follows. Define the baseline distribution $P_{n,s} = (1-s)P_{n,0} + sP_{n,1}$ for some $s \in (0,1)$ and define the distribution neighborhoods

$$\mathcal{U}_\epsilon^{a,D^c}(P_{n,s}) := \{(1-s)P_{n,0} + sQ : D^c(Q\|P_{n,1}) \le \epsilon\}. \tag{105}$$

The corresponding DRO problem can be rewritten as

$$\inf_{\theta \in \Theta} \sup_{Q \in \mathcal{U}_\epsilon^{a,D^c}(P_{n,s})} E_Q[\mathcal{L}_\theta] = \inf_{\theta \in \Theta} \left\{ (1-s)E_{P_{n,0}}[\mathcal{L}_\theta] + s \sup_{Q:D^c(Q\|P_{n,1}) \le \epsilon} E_Q[\mathcal{L}_\theta] \right\}, \tag{106}$$

where one can clearly see that the objective on the right-hand side is non-robust in $P_{n,0}$ but uses OT-regularized-divergence robust optimization for the $P_{n,1}$ component. The parameter $s$ weights the relative importance of the partition components in the overall loss; it can be chosen to correspond to the relative sizes of the partition components (i.e., so that $P_{n,s} = P_n$) or it can be used as a hyperparameter. The supremum over $Q$ on the right-hand side of (106) can be evaluated by the method discussed in Section 2 and the resulting expression is used in what we call the $adv^a$ methods. Similar to the notation in Appendix C.6, the $adv_s^a$ variants refer to the use of adversarial samples constructed via the OT-regularized loss (23) while $adv_{s,l}^a$ refers to the use of adversarial samples together with adversarial labels, both of which are constructed via the OT-regularized loss (25). This method can easily be extended to partitions with more than two components, though we only utilize two components in Section 3.

**Asymmetric Robust Optimization Using a Mixture of Adversarial and Natural Samples:** One can combine asymmetry with the use of natural samples. To do this, choose a mixing parameter $t \in (0,1)$ and define the distribution neighborhoods

$$\mathcal{U}_{\epsilon,t}^{a,D^c}(P_{n,s}) := \{tP_{n,s} + (1-t)((1-s)P_{n,0} + sQ) : D^c(Q\|P_{n,1}) \le \epsilon\} \tag{107}$$
$$= \{(1-s)P_{n,0} + tsP_{n,1} + (1-t)sQ : D^c(Q\|P_{n,1}) \le \epsilon\}.$$

The corresponding DRO problem can be rewritten as

$$\inf_{\theta \in \Theta} \sup_{Q \in \mathcal{U}_{\epsilon,t}^{a,D^c}(P_{n,s})} E_Q[\mathcal{L}_\theta] \tag{108}$$

$$= \inf_{\theta \in \Theta} \left\{ (1-s)E_{P_{n,0}}[\mathcal{L}_\theta] + tsE_{P_{n,1}}[\mathcal{L}_\theta] + (1-t)s \sup_{Q:D^c(Q\|P_{n,1}) \le \epsilon} E_Q[\mathcal{L}_\theta] \right\}.$$

Once again, the supremum over $Q$ on the right-hand side can be evaluated by the method discussed in Section 2 and the resulting expression is used in what we call the $adv^a + nat$ methods.

## C.8 Interpolating between OT-Regularized-$D_f$ and OT Methods

Here we describe a general procedure for modifying an $f$-divergence into a one-parameter family, $D_{f_\beta}$, so that the resulting OT-regularized-$D_{f_\beta}$ method interpolates between OT DRO and OT-regularized-$D_f$ DRO. In particular, this enables us to examine the effect of "turning off" the information divergence component of the method, i.e., the adversarial sample weights (see Section 2.1).

Given an $f$-divergence, $D_f$, and $\beta \in (0,1]$ define

$$f_\beta(z) = \beta f((z-1+\beta)/\beta), \tag{109}$$

(not to be confused with the $\alpha$-divergences, Eq. 29). The $f_\beta$ are convex and $f_\beta(1) = 0$ for all $\beta$, hence $D_{f_\beta}$ is a well-defined family of divergences with $\beta = 1$ giving the original $f$-divergence. It is straightforward to compute the Legendre transform of $f_\beta$ in terms of that of $f$,

$$f_\beta^*(z) = \beta f^*(z) + (1-\beta)z. \tag{110}$$

Therefore one can use $f_\beta$ to define an OT-regularized-divergence DRO problem and simplify it as follows

$$\inf_{\theta \in \Theta} \sup_{Q:D_{f_\beta}^c(Q\|P_n) \le \epsilon} E_Q[\mathcal{L}_\theta] \tag{111}$$

$$= \inf_{\lambda > 0, \rho \in \mathbb{R}, \theta \in \Theta} \left\{ \epsilon\lambda + \beta(\rho + \lambda\frac{1}{n}\sum_{i=1}^n f^*(\mathcal{L}_{\theta,\lambda}^c(x_i) - \rho/\lambda)) + (1-\beta)\lambda\frac{1}{n}\sum_{i=1}^n \mathcal{L}_{\theta,\lambda}^c(x_i) \right\}.$$

As $\beta \to 0^+$ the objective function in (111) approaches that of OT-DRO and so (111) can be thought of as a mixture of OT DRO and OT-regularized-$D_f$ DRO. Moreover, the mixing parameter $\beta$ sets a lower bound on the adversarial sample weights (91), with $p_{*,i} \geq (1 - \beta)/n$. In the KL case one can evaluate the infimum over $\rho$ in (111) to obtain

$$\inf_{\theta \in \Theta} \sup_{Q : KL^c_\beta(Q \| P_n) \leq \epsilon} E_Q[\mathcal{L}_\theta] \tag{112}$$

$$= \inf_{\lambda > 0, \theta \in \Theta} \left\{ \epsilon \lambda + \beta \lambda \log \left( \frac{1}{n} \sum_{i=1}^{n} \exp(\mathcal{L}^c_{\theta,\lambda}(x_i)) \right) + (1 - \beta) \lambda \frac{1}{n} \sum_{i=1}^{n} \mathcal{L}^c_{\theta,\lambda}(x_i) \right\} .$$

When $D_f$ is an $\alpha$-divergence we denote the method (111) by $ARMOR_{\alpha,\beta}$. We denote the method (112) by $ARMOR_{KL_\beta}$. In our tests on MNIST we found that the performance degrades significantly as $\beta$ decreases to 0, both in terms of adversarial robustness or performance generalizability (see Figure 1). This implies that the information divergence, and the adversarial sample weights which it generates, contributes non-trivially to the success of the method.

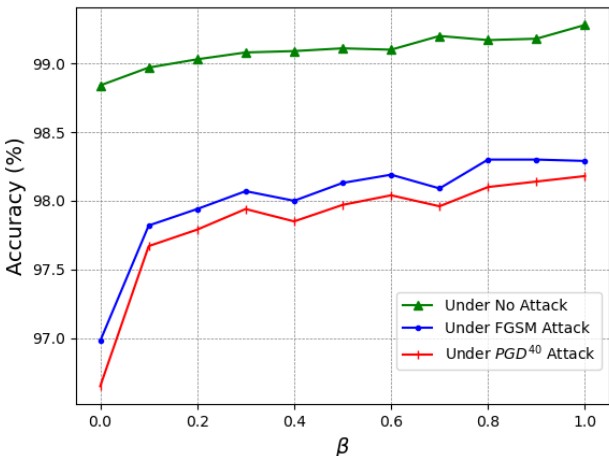

Figure 1: MNIST $ARMOR_{\alpha,\beta}$-robustified model performance against $\beta$, where $\beta = 1$ corresponds $ARMOR_\alpha$ ($adv_{s,l} + nat$).

# D ADDITIONAL EXPERIMENTS

## D.1 COMPARISON OF $ARMOR_D$ VARIANTS

In this section, we present the results of our experiments where $ARMOR_D$'s hyperparameters are tuned with the goal of enhancing performance generalizability on the test set. Table 4 and Table 5 summarize these results on MNIST and the malware dataset, respectively. As shown in Table 4, on MNIST, our proposed $ARMOR_\alpha$ ($adv_s + nat$) achieves the accuracy of $99.30\%$, FNR of $0.70\%$, and FPR of $0.08\%$, outperforming all benchmark methods across all three evaluation metrics. $ARMOR_\alpha$ ($adv_{s,l} + nat$) yields the second best performance generalizability ($99.28\%$ accuracy, $0.73\%$ FNR, and $0.08\%$ FPR). Additionally, it is observed that in almost all cases, the performance under attack is also improved as a result of adversarial training with our proposed method. As observed in Table 5, on the malware dataset, our proposed $ARMOR_\alpha$ ($adv_{s,l}$) achieves the accuracy of $93.9\%$, FNR of $5.19\%$ and FPR of $9.97\%$ outperforming the benchmark methods and the non-robust methods across all three evaluation metrics.

Table 3: **MNIST adversarial training to enhance performance under attack:** Comparison of the performance of our proposed method for enhancing the robustness on the MNIST dataset. Hyperparameters were tuned to enhance performance under attack, though we find that performance when not under attack is simultaneously improved. $adv_s$ denotes the use of adversarial samples constructed via (23) and $adv_{s,l}$ denotes the use of both adversarial samples and labels, as in (25). $nat$ refers to the use of natural samples alongside the adversarial samples, as described in Appendix C.6. See Table 4 for results tuned to enhance performance generalizability.

| | PGD Attack | | | FGSM Attack | | | No Attack | | |
|---|---|---|---|---|---|---|---|---|---|
| **Defense** | Acc | FNR | FPR | Acc | FNR | FPR | Acc | FNR | FPR |
| Non-robust | 25.36% | 74.23% | 8.29% | 52.76% | 46.96% | 5.24% | 99.01% | 1.00% | 0.11% |
| FGSM | 96.63% | 3.39% | 0.37% | 97.45% | 2.57% | 0.28% | 99.05% | 0.96% | 0.11% |
| PGD | 97.10% | 2.92% | 0.32% | 97.26% | 2.76% | 0.30% | 99.15% | 0.85% | 0.09% |
| $ARMOR_{KL}$ $(adv_s)$ | 97.68% | 2.34% | 0.26% | 98.10% | 1.92% | 0.21% | 99.25% | 0.76% | 0.08% |
| $ARMOR_{KL}$ $(adv_s + nat)$ | 98.11% | 1.91% | 0.21% | 98.23% | 1.78% | 0.20% | 99.09% | 0.92% | 0.10% |
| $ARMOR_{\alpha}$ $(adv_s)$ | 97.31% | 2.71% | 0.30% | 97.63% | 2.40% | 0.26% | 99.15% | 0.86% | 0.09% |
| $ARMOR_{\alpha}$ $(adv_s + nat)$ | 98.05% | 1.97% | 0.22% | 98.25% | 1.77% | **0.19%** | **99.30%** | **0.70%** | **0.08%** |
| $ARMOR_{\alpha}$ $(adv_{s,l} + nat)$ | **98.18%** | **1.83%** | **0.20%** | **98.29%** | **1.72%** | **0.19%** | 99.21% | 0.80% | 0.09% |

*Note:* Best metrics are shown in bold font. The numbers for methods that outperform the non-robust model and prior adversarial robustness methods across all three metrics are underlined.

Table 4: **MNIST adversarial training to enhance performance generalizability:** Comparison of the performance of our proposed method on the MNIST dataset where hyperparameters were tuned to enhance performance generalizability without attack. We find that performance when under attack is simultaneously improved. $adv_s$ denotes the use of adversarial samples constructed via (23) and $adv_{s,l}$ denotes the use of both adversarial samples and labels, as in (25). $nat$ refers to the use of natural samples alongside the adversarial samples, as described in Appendix C.6.

| | No Attack | | | PGD Attack | | | FGSM Attack | | |
|---|---|---|---|---|---|---|---|---|---|
| **Defense** | Acc | FNR | FPR | Acc | FNR | FPR | Acc | FNR | FPR |
| Non-robust | 99.01% | 1.00% | 0.11% | 25.36% | 74.23% | 8.29% | 52.76% | 46.96% | 5.24% |
| FGSM | 99.05% | 0.96% | 0.11% | 96.63% | 3.39% | 0.37% | 97.45% | 2.57% | 0.28% |
| PGD | 99.15% | 0.85% | 0.09% | 97.10% | 2.92% | 0.32% | 97.26% | 2.76% | 0.30% |
| $ARMOR_{KL}$ $(adv_s)$ | 99.25% | 0.76% | **0.08%** | 97.68% | 2.34% | 0.26% | 98.10% | 1.92% | 0.21% |
| $ARMOR_{KL}$ $(adv_s + nat)$ | 99.15% | 0.85% | 0.09% | 97.99% | 2.03% | 0.22% | 98.04% | 1.98% | 0.22% |
| $ARMOR_{\alpha}$ $(adv_s)$ | 99.16% | 0.85% | 0.09% | 97.26% | 2.77% | 0.30% | 97.67% | 2.35% | 0.26% |
| $ARMOR_{\alpha}$ $(adv_s + nat)$ | **99.30%** | **0.70%** | **0.08%** | 98.05% | 1.97% | 0.22% | **98.25%** | **1.77%** | **0.19%** |
| $ARMOR_{\alpha}$ $(adv_{s,l} + nat)$ | 99.28% | 0.73% | **0.08%** | **98.07%** | **1.95%** | **0.21%** | 98.17% | 1.85% | 0.20% |

*Note:* Best metrics are shown in bold font. The numbers for methods that outperform the non-robust model and prior adversarial robustness methods across all three metrics are underlined.

Table 5: **Malware adversarial training to enhance performance generalizability:** Comparison of the performance of our proposed method on the malware dataset, Al-Dujaili et al. (2018), where the hyperparameters were tuned to maximize performance generalizability without attack. We find that performance under attack is simultaneously improved. $adv_s$ denotes the use of adversarial samples constructed via (23) and $adv_{s,l}$ denotes the use of both adversarial samples and labels, as in (25). $nat$ refers to the use of natural samples alongside the adversarial samples, as described in Appendix C.6. $adv^a$ refers to asymmetric methods, as described in Appendix C.7, with only the malicious samples robustified.

| | No Attack | | | $rFGSM^k$ Attack | | | Grosse et al. Attack | | |
|---|---|---|---|---|---|---|---|---|---|
| **Defense** | Acc | FNR | FPR | Acc | FNR | FPR | Acc | FNR | FPR |
| Non-robust | 92.96% | 5.30% | 10.13% | 14.71% | 77.85% | 98.48% | 33.03% | 99.86% | 8.53% |
| Grosse et al. | 92.38% | 5.47% | 11.45% | 57.36% | 10.96% | 98.91% | **91.08%** | **8.04%** | 10.48% |
| $rFGSM^k$ (Al-Dujaili et al.) | 92.83% | 5.20% | 10.66% | 60.79% | 4.99% | 100.00% | 74.74% | 32.39% | 12.59% |
| $ARMOR_{KL}$ ($adv_s$) | 93.02% | 5.16% | 10.23% | 61.50% | 6.62% | 95.13% | 84.73% | 22.60% | 2.23% |
| $ARMOR_{KL}$ ($adv_s^a$) | 92.86% | **4.81%** | 11.27% | 68.60% | 10.39% | 68.73% | 82.34% | 19.52% | 14.37% |
| $ARMOR_{KL}$ ($adv_s + nat$) | 92.90% | 5.02% | 10.81% | **84.25%** | 23.33% | **2.28%** | 85.53% | 20.50% | 3.73% |
| $ARMOR_{KL}$ ($adv_s^a + nat$) | 93.02% | 5.39% | **9.82%** | 77.34% | **3.14%** | 57.34% | 72.45% | 35.12% | 14.11% |
| $ARMOR_{\alpha}$ ($adv_s$) | 92.92% | 5.09% | 10.63% | 83.23% | 5.60% | 36.62% | 89.26% | 9.67% | 12.64% |
| $ARMOR_{\alpha}$ ($adv_s^a$) | 92.90% | 4.86% | 11.09% | 68.93% | 9.26% | 69.82% | 85.20% | 15.89% | 12.87% |
| $ARMOR_{\alpha}$ ($adv_s + nat$) | 93.04% | 5.04% | 10.36% | 64.73% | 5.69% | 87.82% | 86.49% | 20.15% | **1.73%** |
| $ARMOR_{\alpha}$ ($adv_s^a + nat$) | 92.84% | 4.99% | 11.02% | 70.90% | 5.44% | 71.12% | 59.05% | 55.11% | 15.81% |
| $ARMOR_{\alpha}$ ($adv_{s,l}$) | **93.09%** | 5.19% | 9.97% | 59.52% | 8.72% | 96.90% | 72.68% | 36.86% | 10.38% |

*Note:* Best metrics are shown in bold font. The numbers for methods that outperform the non-robust model and prior adversarial robustness methods across all three metrics are underlined.

## D.2    FURTHER COMPARATIVE ANALYSIS OF THE PROPOSED METHOD

To further illustrate the effectiveness of the proposed method, we examine a subset of MNIST test digits for which the performance of the FGSM, PGD, and our $ARMOR_{\alpha}$ ($adv_{s,l} + nat$) methods differ, i.e., the digits for which at least one, but not all three of these methods failed under adversarial attack. This subset consists of 225 digits out of the 10,000 digit MNIST test set. The models robustified with FGSM, PGD, and our method correctly classified 54 digits (24.00%), 100 digits (44.44%), and 198 digits (88.00%) out of this set, respectively. Figure 2 shows the results on 12 randomly selected examples from the constructed set of 225 digits. We see that our method exhibits significantly greater robustness on this subset of digits, and on the test set overall (see Table 3), though there remains a subset of "difficult" samples on which all three methods fail under adversarial attack. Figure 2a depicts the performance of the non-robust model under the PGD attack for the random sample described above, while Figure 2b, Figure 2c, and Figure 2d show the corresponding performance under PDG attack of the same CNN model robustified by adversarial training with FGSM, PDG, and our proposed $ARMOR_{\alpha}$ ($adv_{s,l} + nat$) method respectively. All attacks in this section were conducted by the neighborhood size of 0.1, $\ell_{\infty}$ neighborhood, 20 iterations for adversarial training, and 40 iterations for evaluation of adversarial robustness.

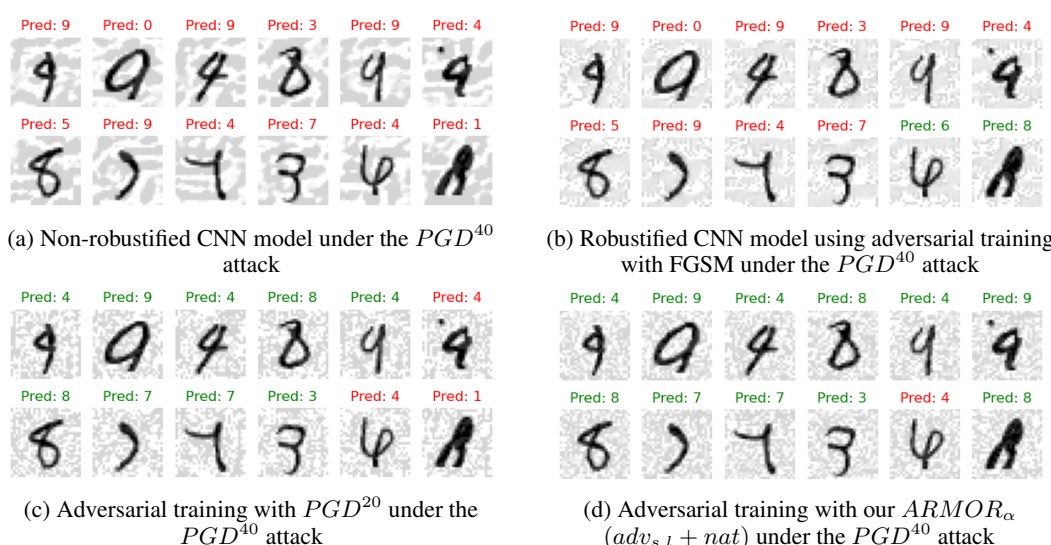

(a) Non-robustified CNN model under the $PGD^{40}$ attack

(b) Robustified CNN model using adversarial training with FGSM under the $PGD^{40}$ attack

(c) Adversarial training with $PGD^{20}$ under the $PGD^{40}$ attack

(d) Adversarial training with our $ARMOR_\alpha$ $(adv_{s,l} + nat)$ under the $PGD^{40}$ attack

Figure 2: The majority of the images, all except the second digit in the bottom row ("7"), were classified correctly by the non-robust CNN model when not under attack, but all 12 samples modified with the $PGD^{40}$ attack successfully mislead the non-robust CNN (Figure 2a). The robustified model via adversarial training with FGSM leads to two correct predictions (for digits "6" and "8") (Figure 2b). The number of correct predictions in this sample increases to 9 via adversarial training with $PGD^{20}$ (Figure 2c). The number of correct predictions increases to 11 with our proposed method under the same attack (Figure 2d).

