# OpenReview forum: "Adversarially Robust Deep Learning with Optimal-Transport-Regularized Divergences"
_ICLR.cc/2024/Conference — Submitted to ICLR 2024_

### Official Review · Reviewer_8rju · 2023-10-15

**Soundness:** 2 fair
**Presentation:** 2 fair
**Contribution:** 2 fair
**Rating:** 3
**Confidence:** 4

**Summary:**

This paper defines a new divergence considering both the information divergence and optimal transport and then investigates adversarial robustness in terms of the defined divergence from a distributionally robust optimization perspective. Based on the analysis, the authors propose the ARMOR\_D algorithm and then obtain SOTA performance on MNIST and Malware compared with FGSM, PGD, TRADES, and MART.

**Strengths:**

- The paper generalizes the adversarial training framework and OT-DRO by defining a new divergence considering both the information divergence and optimal transport.
- The paper provides a theoretical analysis of the proposed framework.

**Weaknesses:**

- The empirical evaluation is not sufficient, which is not enough to show the effectiveness of the proposed method.
  - The baselines are weak. The latest baseline used in the paper is proposed in 2020. However, it is 2023 now, there are so many good methods proposed these years. So I think the authors should compare their methods with the latest SOTA method.
  - There are only **two small** datasets. Firstly, I think the authors should conduct experiments on more datasets to indicate consistent improvement. Secondly, the experiments should be conducted on larger datasets, MNIST is a very easy task, the authors should conduct experiments on CIFAR10, CIFAR100, and ImageNet.
  - The backbone is small, which is not consistent with the word "deep learning" in the topic of this paper. Larger neural networks should be used (together with larger datasets).
  - Stronger attack methods should be used. The paper only reports performance under FGSM and PGD attacks, which are not strong enough. Stronger attacks such as CW and AutoAttack should be used.
- There are some other works that investigate adversarial robustness in terms of distributionally robust optimization [1-2], the discussions of related works are mission. Furthermore, given the existence of these papers, I think this paper is not novel enough.
- The theoretical analyses only show that the proposed framework is more general, but do not show why the more general framework behaves better. In general, a more specific framework may contain additional knowledge about the task and lead to better performance, so the fact that a more general framework leads to better performance is strange to me. I think it needs to be carefully explained.


### references

[1] Certifying Some Distributional Robustness with Principled Adversarial Training.

[2] A Distributional Robustness Perspective on Adversarial Training with the $\infty$-Wasserstein Distance.

**Questions:**

- See the weaknesses.
- Why adding natural examples to the training can improve adversarial robustness? Does this still hold for larger datasets such as the CIFAR and the ImageNet?

---

> ### Author Response · Authors · 2023-11-22
>
> "The baselines are weak. The latest baseline used in the paper is proposed in 2020. However, it is 2023 now, there are so many good methods proposed these years. So I think the authors should compare their methods with the latest SOTA method. "
>
> - In the revised manuscript, we reviewed, cited, and compared our method with two additional new methods, one of which was pointed by you: Regniez et al. 2022 (A Distributional Robustness Perspective on Adversarial Training with the $\infty$-Wasserstein Distance). We also found a more recent work by Bui et al., 2022 (A Unified Wasserstein Distributional Robustness Framework for Adversarial Training) published in ICLR22 that we used to compare our performance on the much stronger Auto-attack and PGD-200. In addition, we modified  the MNIST example to use the standard PGD-AT inner maximizer. This focuses the preliminary MNIST example on    highlighting the unique feature of our method: the  principled addition of adversarial sample re-weighting (based on an f-divergence cost) on top of adversarially modified samples. We find it significantly improves the performance under AutoAttack compared to the baseline as well as to the recent method Bui et al. 2022, which is another method that can be used to augment any empirical risk minimization problem.  After highlighting the effect of re-weighting alone, we move on to the higher-dimensional and more challenging malware example where we also experiment with modifying the OT cost on top of the adversarial re-weighting.
>
>
> "There are only two small datasets. Firstly, I think the authors should conduct experiments on more datasets to indicate consistent improvement. Secondly, the experiments should be conducted on larger datasets, MNIST is a very easy task, the authors should conduct experiments on CIFAR10, CIFAR100, and ImageNet."
>
> - For the discrete case, our dataset is high-dimensional (22,761 features), which is higher than CIFAR10 (3,072). To converge the new experiments on time, we dedicated our limited  resources to benchmarking against Auto-attack.
>
> "The backbone is small, which is not consistent with the word "deep learning" in the topic of this paper. Larger neural networks should be used (together with larger datasets)."
>
> - To not emphasize on the word ``deep," we removed the word from the title as well as the name of the proposed method.
>
> "Stronger attack methods should be used. The paper only reports performance under FGSM and PGD attacks, which are not strong enough. Stronger attacks such as CW and AutoAttack should be used."
>
> - We added evaluations on Auto-attack as a comprehensive and strong attack that is an ensemble of four newer attacks based on your comment.
>
> "There are some other works that investigate adversarial robustness in terms of distributionally robust optimization [1-2], the discussions of related works are mission. Furthermore, given the existence of these papers, I think this paper is not novel enough.      The theoretical analyses only show that the proposed framework is more general, but do not show why the more general framework behaves better. In general, a more specific framework may contain additional knowledge about the task and lead to better performance, so the fact that a more general framework leads to better performance is strange to me. I think it needs to be carefully explained."
>
> - We added Section 2.1, which explains the fundamental difference between our method and prior distributionally robust optimization based methods.  Specifically, we show that in addition to adversarially transporting samples (as in prior methods) our approach dynamically re-weights samples via f-divergence to focus the optimization towards improving performance on the more troublesome samples; see Eq. 16-17.  Conceptually, this is the key  innovation of our method, and is not present in prior DRO-based approaches.
>
> "Why adding natural examples to the training can improve adversarial robustness? Does this still hold for larger datasets such as the CIFAR and the ImageNet?"
>
> - The sample re-weighting component of our method focuses the optimizer towards the most troublesome adversarial samples.  When using alpha-divergences, some adversarial samples in each minibatch can even be completely ignored if their loss is small enough.  However ignoring natural samples can be a problem, especially at the beginning of training, before the model has learned anything.  We find that combining the natural loss, which incorporates all samples in each minibatch, with the adversarial loss, which focusing on the most difficult samples,  generally provides the best performance.

---

### Official Review · Reviewer_sgR7 · 2023-10-28

**Soundness:** 4 excellent
**Presentation:** 3 good
**Contribution:** 3 good
**Rating:** 5
**Confidence:** 3

**Summary:**

The paper provides a novel method for adversarially robust training of deep learning models. Essentially, authors propose to use the extension of the common Adversarial Training where the internal maximization process should be conducted on top of both 1) divergence term between the "close" adversarial distribution and the original empirical one, and 2) optimal transport between these two divergences.

**Strengths:**

First of all, a lot of seeming correctly theoretical background and proofs are provided (especially in the Appendix A), which make the work standing out of a common adversarial robustness methods.
Additionally, authors provided a rigorous experimentation analysis of different variants of their method for two modalities (images and malware detection) and prove the superiority of their approach.
Finally, which is very nice, their method improves even no-attack setting, which is very important when we don't have adversaries - the system still should work reliable.

**Weaknesses:**

Although the paper is very well written with a lot of theoretical details, there are still some (I hope minor) weaknesses:
- the work used a super small and toy dataset for images - MNIST - the scale and reliability on results obtained working with it are unlikely used anywhere in reality. Moreover, the conclusion obtained can be misleading. The golden standard is to use ImageNet as a must for the image datasets (probably the same is for the Malware detection dataset, I have just not worked with it)
- when comparing the results shown in Tables 1 and 2, we can see that the best OT-regularized divergences methods in terms of accuracy are very different for image recognition and malware detection problem. What is the root cause behind it? No any written hypothesis or discussion. I guess (and see the item above) it is somehow related to MNIST dataset and that the results there are like 98-99% which makes all the improvement marginal and not very generalizable
- starting the Page 2, the cost function $c(x,y)$ is introduced, but all the theorems and results are only valid if $c(x,y)$ is non-negative, but it is not mentioned in the main text of the paper (only in Appendix)

**Questions:**

Here are a couple of questions
- what was the original motivation to combine OT and DRO? E.g., some clear bad cases for OT are not in the common DRO and vice versa etc.
- what was the reasoning behind choosing a probability-related term for the cost function as a $g_{\delta}(z) = z/(1-z/\delta)$? To mimic sigmoid?

---

> ### Author Response · Authors · 2023-11-22
>
> "The work used a super small and toy dataset for images - MNIST - the scale and reliability on results obtained working with it are unlikely used anywhere in reality. Moreover, the conclusion obtained can be misleading. The golden standard is to use ImageNet as a must for the image datasets (probably the same is for the Malware detection dataset, I have just not worked with it)"
>
> - We agree that for the continuous case, testing on higher-dimensional datasets can be beneficial. To converge the new experiments on time, we dedicated our limited  resources to benchmarking against Auto-attack. We also note that our malware dataset, also used to demonstrate the effectiveness of the model in discrete cases, has 22,761-dimensional feature vectors, which is higher than CIFAR10 (3,072 features) and on par with ImageNet (23,296 features).
>
> "when comparing the results shown in Tables 1 and 2, we can see that the best OT-regularized divergences methods in terms of accuracy are very different for image recognition and malware detection problem. What is the root cause behind it? No any written hypothesis or discussion. I guess (and see the item above) it is somehow related to MNIST dataset and that the results there are like 98-99\% which makes all the improvement marginal and not very generalizable"
>
> - In our new experiments, we are comparing against much stronger attacks with increased epsilon ball, including many more PGD steps (PGD-200) and the  much stronger  AutoAttack. Accordingly, the new results are not in the 98-99 neighborhood any more.  In addition to including much stronger attacks,  we also modified the MNIST example to use the standard PGD-AT inner maximizer. This focuses the preliminary MNIST example on    highlighting the unique feature of our method: the  principled addition of adversarial sample re-weighting (based on an f-divergence cost) on top of adversarially modified samples. We find it significantly improves the performance under AutoAttack compared to the baseline as well as to the recent method Bui et al. 2022 (published in ICLR22) which is another method that can be used to augment any empirical risk minimization problem.  After highlighting the effect of re-weighting alone, we move on to the higher-dimensional and more challenging malware example where we also experiment with modifying the OT cost on top of the adversarial re-weighting.
>
> "starting the Page 2, the cost function is introduced, but all the theorems and results are only valid if is non-negative, but it is not mentioned in the main text of the paper (only in Appendix)"
>
> - We added a line stating the assumptions on $c$ after equation 4 in the text.
>
> "what was the original motivation to combine OT and DRO? E.g., some clear bad cases for OT are not in the common DRO and vice versa etc."
>
> - OT allows samples to be perturbed (transported) into adversarial samples.  The $f$-divergence component of the DRO framework allows samples to be re-weighted. The motivation was that the re-weighting will shift weight towards the more troublesome samples, thereby making the optimizer focus on the more troublesome samples.  We included a more detailed discussion of this intuition in the new Section 2.1.
>
> "what was the reasoning behind choosing a probability-related term for the cost function as a $g_\delta(z)=z/(1-z/\delta)$? To mimic sigmoid?"
>
> - In essence, we wanted a function that is smooth, increasing, has a vertical asymptote, and is easy to compute (similar to the behavior of the inverse of sigmoid). The output of the classifier network can be viewed as a probability vector, so it seemed plausible that giving the adversarial sample/label pair this same structure could improve the regularizing effects of a DRO-based method. The structure of the OT cost was chosen to include a vertical asymptote in $g_\delta$, to ensure that the optimization results in an adverarial probability vector that still predicted the original class label; the difference is only that the certain label is relaxed to a (less certain) probability vector.  We added some clarification of this point to the paragraph above eq. 25.

---

> > ### Comment · Reviewer_sgR7 · 2023-11-23
> >
> > Thanks for answering and addressing my comments. Increasing the grade.

---

### Official Review · Reviewer_4TT9 · 2023-10-31

**Soundness:** 3 good
**Presentation:** 3 good
**Contribution:** 3 good
**Rating:** 5
**Confidence:** 4

**Summary:**

This paper introduces the novel approaches to enhancing the adversarial robustness of deep learning models. This paper enhances adversarial robustness by maximizing the expected loss over a neighborhood of distributions for distributionally robust optimization. For constructing adversarial samples, the proposed method allows samples to be both transported and re-weighted, according to the OT cost and the information divergence. The authors demonstrate the effectiveness of the proposed method on malware detection and image recognition applications and find that it outperforms existing methods at enhancing the robustness against adversarial attacks.

**Strengths:**

1. This work proposes a novel class of divergences for comparing probability distributions called optimal-transport-regularized divergences and uses them to construct distribution neighborhoods for use in distributionally robust optimization.
2. This paper proves a number of new properties of the OT-regularized divergences and demonstrates the effectiveness of the OT-regularized divergence method as a novel approach to enhancing adversarial robustness in deep learning leading to substantial performance gains.

**Weaknesses:**

1. For the classification problem on the continuous data, this paper tests the ARMORD adversarial robustness methods on MNIST digit classification, where the dimension of image in the MNIST is 28*28. Note that the computation of Optimal Transport (OT) is closely related to the dimension of data, thus it would be interesting to see the performance of the proposed approach and its counterparts on the image data with larger dimension such as CIFAR-10 with 3*32*32 or the tiny-ImageNet with 3*64*64.
2. In the comparison of the performance for enhancing the robustness on the MNIST dataset, only the gradient-based attacks are used to evaluate the adversarial robustness. It would be more sufficient for the adversarial robustness of proposed Optimal-Transport-Regularized Divergences if there are evaluations of the optimization-based attack such as CW and the stronger comprehensive attack AutoAttack.
3. Note that the proposed OT-regularized divergences may need more computational cost than the conventional divergence, thus it would be more meaningful for the practical use to provide the comparison of the computational costs between the proposed OT-regularized divergences based adversarial training and the counterparts.

**Questions:**

This paper proposes the optimal-transport-regularized divergences to enhance the adversarial robustness of deep learning models. Yet, it still needs more details and evidences to further explain the effectiveness of the proposed approach.

---

> ### Author Response · Authors · 2023-11-22
>
> "For the classification problem on the continuous data, this paper tests the ARMORD adversarial robustness methods on MNIST digit classification, where the dimension of image in the MNIST is 2828. Note that the computation of Optimal Transport (OT) is closely related to the dimension of data, thus it would be interesting to see the performance of the proposed approach and its counterparts on the image data with larger dimension such as CIFAR-10 with 33232 or the tiny-ImageNet with 364*64."
>
> - We agree that similar to our high-dimensional discrete dataset, the continuous case can also benefit from testing on higher-dimensional datasets. To converge the new experiments on time, we dedicated our limited  resources to benchmarking against Auto-attack. Regarding the computational cost of OT, it is only the OT cost function that occurs in the inner maximizer, which is relatively easy to compute. Our method doesn't require the computation of the optimal coupling, which would be much more difficult.  This is clarified in our derivation, Eq.6-9.  We also want to point out that our method can still use simple OT cost function (Eq. 21) that leads to the same PGD-AT inner maximizer as Madry et al. 2018, while retaining the novel adversarial re-weighting  provided by the f-divergence component of our method;  proposing a unified framework to incorporate these two mechanisms in a consistent and principled manner is the primary innovation of our method which, to our knowledge, has not been studied in the adversarial robustness literature.  The unique features are further discussed in the new Section 2.1. To complement this discussion, we modified the MNIST example in Section 3.1 to use the standard PGD-AT inner maximizer, thus focusing this preliminary example on highlighting the unique feature of our method: the combination of adversarial sample re-weighting (based on an f-divergence cost).  This is an important test, before we move on to the higher-dimensional malware example where we also experiment with modifying the OT cost on top of the adversarial re-weighting.
>
> "In the comparison of the performance for enhancing the robustness on the MNIST dataset, only the gradient-based attacks are used to evaluate the adversarial robustness. It would be more sufficient for the adversarial robustness of proposed Optimal-Transport-Regularized Divergences if there are evaluations of the optimization-based attack such as CW and the stronger comprehensive attack AutoAttack."
>
> - Thanks for directing us to Auto-attack as a comprehensive attack incorporating a set of four strong attacks. We conducted a new set of experiments to evaluate our performance against auto-attack and compare with recent adversarial robustness methods. see Table 1.
>
> "Note that the proposed OT-regularized divergences may need more computational cost than the conventional divergence, thus it would be more meaningful for the practical use to provide the comparison of the computational costs between the proposed OT-regularized divergences based adversarial training and the counterparts."
>
> - The additional computation cost is relatively minimal. In the inner maximizer there is an additional term involving the OT cost function, though one can even choose that be a simple hard constraint as in Eq. 21 which requires no additional derivatives to be computed in the inner maximizer.  In the outer minimizer, the f-divergence component of the method is seen in the function $f^*$, which is tantamount to  adding one additional activation function layer (1D).  And then there are the two additional real parameters $\lambda,\rho$ that are appended to the NN parameters. Overall these are minimal additions, considering the number and size of the layers. We also note that our method does not require the computation of the optimal coupling (a much more costly task), rather it simply uses the optimal transport cost function in the inner maximizer.
>
> "This paper proposes the optimal-transport-regularized divergences to enhance the adversarial robustness of deep learning models. Yet, it still needs more details and evidences to further explain the effectiveness of the proposed approach."
>
> - We added Section 2.1, which explains the fundamental difference between our method and prior distributionally robust optimization based methods.  Specifically, we show that in addition to adversarially transporting samples (as in prior methods) our approach dynamically re-weights samples via f-divergence to focus the optimization towards improving performance on the more troublesome samples; see Eq. 16-17.  Conceptually, this is the key  innovation of our method, and is not present in prior DRO-based approaches.

---

### Official Review · Reviewer_2esQ · 2023-11-06

**Soundness:** 3 good
**Presentation:** 3 good
**Contribution:** 3 good
**Rating:** 5
**Confidence:** 4

**Summary:**

This paper proposed adversarial training with optimal-transport-regularized divergence. It is a framework solving distributional robustness optimization problem where the distributional divergence is regularized by optimal transport. The authors comprehensively study the properties of this problem and demonstrate the effectiveness of the derived algorithms.

**Strengths:**

The paper proposes a framework that is generally applicable and mathematically elegant. The properties of the OT-regularized divergence are nice and the derived algorithm is nice. The paper is well-written and easy to read.

**Weaknesses:**

The theoretical part is nice, my major concern is the empirical part.

1. The datasets studied in the empirical study are very small and are generally considered as toy examples. Experiments on a larger dataset will make the empirical results more convincing.

2. The robustness evaluation is more comprehensive. For example, PGD and FGSM cannot comprehensively and reliably evaluate the robustness of a model as indicated in [A]. The authors should use auto-attack, which is an ensemble of four different attacks, to comprehensively evaluate the robustness and to make the results more convincing.

[A] "Reliable evaluation of adversarial robustness with an ensemble of diverse parameter-free attacks". Francesco Croce, Matthias Hein.
ICML 2020.

**Questions:**

The paper is generally well-written and interesting. My questions are the two points in the weakness part. The manuscript will be better with the empirical issues addressed.

---

> ### Author Response · Authors · 2023-11-22
>
> "The datasets studied in the empirical study are very small and are generally considered as toy examples. Experiments on a larger dataset will make the empirical results more convincing."
>
> - We agree that for the continuous case, testing on higher-dimensional datasets can be beneficial. To converge the new experiments on time, we dedicated our limited  resources to benchmarking against Auto-attack. We also note that our malware dataset, also used to demonstrate the effectiveness of the model in discrete cases, has 22,761-dimensional feature vectors which is higher than CIFAR10 (3,072 features) and on par with ImageNet (23,296 features).
>
> "The robustness evaluation is more comprehensive. For example, PGD and FGSM cannot comprehensively and reliably evaluate the robustness of a model as indicated in [A]. The authors should use auto-attack, which is an ensemble of four different attacks, to comprehensively evaluate the robustness and to make the results more convincing."
>
> - Thanks for your suggestion regarding adding Autoattack. We added Auto-attack to our experiments, along with changing to the more difficult PGD200 (instead of PGD40). We have also changed the variant of our method used there to more directly highlight the contribution of our method's main innovation (i.e., incorporating adversarial sample re-weighting via f-divergence).  Specifically, we changed the MNIST experiment to use the same PGD-AT inner maximizer as Madry et al., 2018, thereby focusing the example on highlighting the unique feature of our method: the combination of adversarial sample re-weighting (based on an f-divergence cost) with the more commonly used adversarially transported samples. By using the  PGD-AT when constructing adversarial samples, this example now  isolates the contribution of our novel adversarial re-weighting to the performance, providing an important preliminary test example, before we move on to the higher-dimensional malware example, where we also experiment with modifying the OT cost on top of the adversarial re-weighting.     Modifying the MNIST example in this way also serves to complement the discussion in the newly added Section 2.1 (portions of which used to be in an appendix), where we show precisely how our method dynamically re-weights samples to focus the optimizer towards the more troublesome adversarial examples; see especially Eq. 16-17.

---

### Meta-Review · Area_Chair_iUUE · 2023-12-06

**Metareview:**

This paper proposes to tackle the adversarial robustness problem by connecting it to distributional robustness and using a regularized optimal transport formulation. The main strength of this paper is the mathematically grounded motivation of the paper.  The main weakness is the experimental evaluation of the method, that is significantly below the standards of ICLR.

I agree with reviewer sgR7 that the finding that the method doesn't has a negative impact on the standard accuracy is very interesting but should be validated on well established benchmarks (at least CIFAR) since there is not much accuracy-robustness tradeoff on MNIST.

**Justification For Why Not Higher Score:**

The experimental evaluation has been unanimously considered under the ICLR standard.

**Justification For Why Not Lower Score:**

N/A

---

### Decision · Program_Chairs · 2024-01-16

Reject